# CerCE: Towards Certifiable Continual Learning

## Abstract

Continual Learning (CL) aims to develop models capable of learning sequentially without catastrophic forgetting of previous tasks. However, most existing approaches rely on heuristics and lack formal guarantees, limiting their applicability in safety-critical domains. We introduce **Cer**tifiable **C**ontinual **LE**arning (CerCE), a CL framework that provides provable certificates of non-forgetting during training. CerCE leverages Linear Relaxation Perturbation Analysis (LiRPA) to reinterpret weight updates as structured perturbations, deriving constraints that guarantee the preservation of past knowledge. We formulate CL as a constrained optimization problem and propose practical optimization strategies, including gradient projection and Lagrangian relaxation, to efficiently satisfy these certification constraints. Furthermore, we connect our approach to PAC-Bayesian generalization theory, showing that CerCE naturally leads to tighter generalization bounds and reduced memory overfitting. Experiments on standard benchmarks and safety-critical datasets demonstrate that CerCE achieves strong empirical performance while uniquely offering formal guarantees of knowledge retention, marking a significant step toward verifiable continual learning for real-world applications.

## 1 Introduction

Continual learning (CL) seeks to develop machine learning models with the ability to learn from a sequence of tasks without catastrophic forgetting, the tendency of neural networks to lose previously acquired knowledge when trained on new data. While CL has made impressive empirical progress through strategies like regularization, architectural expansion, and experience replay, it remains a largely heuristic domain, lacking strong theoretical guarantees. This is particularly concerning in safety-critical domains such as healthcare and autonomous systems, where forgetting could have detrimental consequences.

A central challenge in CL is that model updates for new tasks often alter parameters critical to performance on previous tasks. Although various methods attempt to mitigate forgetting, such as Elastic Weight Consolidation (EWC) (Kirkpatrick et al., 2017) and Learning without Forgetting (LwF) (Li & Hoiem, 2017), these approaches lack formal guarantees that prior knowledge is preserved. Recent work, such as InterContiNet (Wołczyk et al., 2022), has proposed using weight intervals to prevent forgetting via set intersection. However, such methods severely constrain model capacity, limiting their practical utility and performance.

In this paper, we introduce *Certifiable Continual LEarning (CerCE)* and address a critical need for continual learning: learning procedures that are accompanied by formal guarantees that previously learned examples are not forgotten. We introduce a novel framework grounded in Linear Relaxation Perturbation Analysis (LiRPA), which provides provable bounds on neural network outputs under perturbations to the inputs or model parameters. By interpreting weight updates during training as structured weight perturbations, we can use LiRPA to derive constraints ensuring that previously seen examples remain correctly classified, thereby achieving certifiable non-forgetting (Fig. 1).

The significance of such guarantees goes beyond mitigating forgetting. From an optimization point of view, while existing training methods result in a single parameter point, CerCE provides a neighborhood of parameters where the certificate is guaranteed to hold. If a standard training approach leads to a local minimum where some critical samples remain misclassified, it is non-trivial

to improve the final weights further. On the other hand, CerCE yields a neighborhood of parameters with guaranteed performance that can be sampled for further training.

Finally, we connect our framework to PAC-Bayes generalization theory and show that satisfying the CerCE constraints not only provides certificates of non-forgetting but also leads to tighter generalization bounds. This addresses another important challenge in CL: memory overfitting when using small replay buffers (Zhang et al., 2022).

In summary, our contributions are as follows:

- We propose CerCE, the first continual learning framework that provides certificates of non-forgetting during training via weight perturbation analysis, which is essential for safety-critical systems.
- We formulate continual learning as a constrained optimization problem, where constraints derived from LiRPA guarantee classification accuracy on past data.
- We develop practical optimization strategies that incorporate these constraints efficiently, including gradient projection and Lagrangian optimization.
- We show a connection between CerCE, and tighter PAC-Bayes generalization bounds, reducing overfitting.
- We demonstrate that our approach achieves strong empirical performance on standard benchmarks and safety-critical datasets, while offering theoretical guarantees previously absent from CL methods.

CerCE lays the groundwork for a new class of theoretically grounded continual learning methods that go beyond heuristics and provide robust, certifiable learning dynamics, which are crucial to the development and deployment of continual learning algorithms for safety-critical applications.

## 2 BACKGROUND AND RELATED WORKS

**Notation** Let $\mathcal{D} := (\mathcal{X}, \mathcal{Y})$ be a data distribution of input/output pairs, where $\mathcal{X} \in \mathbb{R}^d$, $\mathcal{Y}$ is drawn from a set of labels, and $\mathcal{S}$ is a set of samples drawn from the distribution $\mathcal{D}$. Let $f_\theta : \mathbb{R}^d \to \mathbb{R}^c$ be a neural network parameterized by $\theta = \{\theta^{(1)}, ..., \theta^{(\omega)}\}$, where $c$ is the number of different classes, and $K$ is the total number of model parameters where in a slight abuse of notation, we denote $\theta \in \mathbb{R}^K$ as the vector of all model parameters. We denote $l_\mathcal{D}(\theta)$ as the *expected* 0-1 error induced on the distribution $\mathcal{D}$ by parameters $\theta$. Subsequently, $\hat{l}_\mathcal{S}(\theta)$ denotes the *empirical* 0-1 error induced on the set $\mathcal{S}$ by $\theta$. We will use $\mathcal{L}(\cdot, \cdot)$ to denote the *cross-entropy* loss function. Note that $l$ is the 0-1 error, i.e. number of misclassified samples,

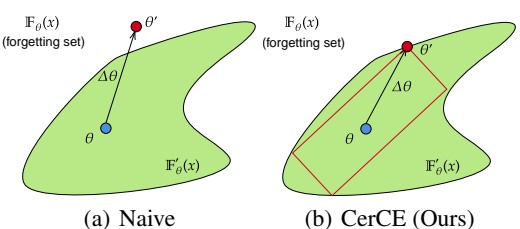

(a) Naive      (b) CerCE (Ours)

Figure 1: Conceptual depiction of the model parameter space. We restrict the model updates to a linear under-approximation of the non-forgetting set.

and thus, different than $\mathcal{L}$. Let $\mathcal{T}_i$, $i \in \mathbb{N}$ be the distribution of a task and $\mathcal{T}_{0:k}$ be a distribution where a sample drawn from $\mathcal{T}_{0:k}$ is equally likely to be drawn from $\mathcal{T}_i$ for any $0 \leq i \leq k$. $\mathcal{N}(\mu, \Sigma)$ refers to the Gaussian distribution with mean vector $\mu$ and covariance matrix $\Sigma$, and $KL(., .)$ is the Kullback–Leibler divergence between two distributions.

**Continual Learning** Existing CL methods can generally be categorized as regularization-based, architecture-based, or rehearsal-based approaches. Architecture-based methods (Rusu et al., 2016; Mallya & Lazebnik, 2018; Wang et al., 2020) modify the model architecture by compartmentalizing and expanding the network parameters, keeping the information from new tasks from interfering catastrophically with the previously learned tasks. Rehearsal-based methods keep a memory buffer of data during training and *replay* these buffered samples to prevent the model from forgetting previous task knowledge. Rehearsal offers a simple and powerful framework to tackle CL and is leveraged in many state-of-the-art methods (Chaudhry et al., 2019; Buzzega et al., 2020; Yoo et al., 2024).

Moreover, an alternative category of CL methods called regularization-based methods (Kirkpatrick et al., 2017; Zenke et al., 2017; Li & Hoiem, 2017; Chaudhry et al., 2018) seeks to ensure model stability through regularization on the weights during optimization. Such methods mainly focus on identifying model weights that are more important to the performance of previous tasks, and impose a penalty on changes to those weights.

Theoretically certifiable continual learning is largely unexplored. InterContiNet (Wołczyk et al., 2022), propose to replace neural network weights with weight intervals, and to take the intersection of the resulting intervals for each task, guaranteeing the worst-case performance of the network does not get worse. However, this approach restricts the learning capabilities of neural networks significantly. Moreover, their guarantee of performance only holds on the worst-case performance of the weight intervals, while evaluation is made on interval centers, leaving room for fluctuations in accuracy. Finally, to the best of our knowledge, Wołczyk et al. (2022) is the only work on the subject of formal guarantees for continual learning. In this paper, and for the first time, we provide a framework as well as a concrete methodology using linear relaxations of neural networks to perform certifiable continual learning.

**Linear Relaxation based Perturbation Analysis (LiRPA)**  LiRPA (Xu et al., 2020; Wang et al., 2021; Zhang et al., 2018; Xu et al., 2021) is a set of methods that use linear relaxations of activation functions to derive provable bounds on neural network outputs under perturbations to nodes of a computational graph. At a high level, given a set of initial permissible perturbations to a neural network input or parameters, LiRPA offers upper and lower bounds to the network output in the form of *linear functions of the input or parameters.* It is worth noting that these bounds are not approximations, and are rather abstractions, and do hold in practice. Typically, perturbations are applied only to the input of the neural network. In contrast, in this work, we analyze perturbations to the weights of the neural network. By re-interpreting weight updates during training as weight perturbations, we can apply LiRPA directly to continual learning to certifiably prevent forgetting by bounding the change in the network output. For a more detailed description of LiRPA, specifically auto-LiRPA (Xu et al., 2020), see Section B. Typically, LiRPA methods are used within the context of adversarial robustness and input perturbations (Zhang et al., 2018; Xu et al., 2021). However, in this work, we view and utilize LiRPA in the much-unexplored context of weight perturbations to perform continual learning. To the best of our knowledge, Weng et al. (2020) is the only other work that studies the certified robustness of feedforward networks under weight perturbations (other than the one experiment in Xu et al. (2020)). In this work, we do not solely study robustness against weight perturbation, but rather use it as a tool to provide guarantees of non-forgetting during CL.

**PAC-Bayes Generalization Bounds**  PAC-Bayes bounds (McAllester, 1999; Pentina & Lampert, 2014; Pérez-Ortiz et al., 2021) are a set of upper bounds on the test error of learning algorithms, including neural networks. In this paper, we introduce CerCE to perform neural network training while providing certificates on non-forgetting. Moreover, we will show that CerCE may result in less *memory-overfitting* –the tendency of overfitting on buffer samples in rehearsal CL– using PAC-Bayes generalization bounds. There is an extensive family of PAC-Bayes bounds (Alquier et al., 2024), and so for the sake of brevity, we will work with a simple and commonly used version of the PAC-Bayes bound as per McAllester (1999). We refer the reader to Section C for further discussion of PAC-Bayes bounds as well as a statement of the generic PAC-Bayes bounds which we use further in our proofs.

## 3 PROBLEM FORMULATION

In this work, we are interested in providing a way to perform continual learning while providing certificates to prevent the forgetting of samples. In this context, a certificate is a theoretical guarantee that given a set of assumptions (e.g., being within some radius of some parameter point $\theta$), a certain result will hold (e.g., a sample $X$ is classified correctly).

**Definition 1** (Forgetting Set). *Given parameters $\theta$ and input sample $x \in \mathbb{R}^d$ with label $y$, we define the **forgetting set** $\mathbb{F}_\theta(x) := \left\{ \Delta\theta \mid f_{\theta+\Delta\theta}(x) \neq y \right\}$ where $y$ is the label of $x$. Subsequently, $\mathbb{F}'_\theta(x)$ is the complement of $\mathbb{F}_\theta(x)$, where the samples are **not** forgotten. Moreover, for a set of samples $X$, let $\mathbb{F}_\theta(X)$ be the intersection of $\mathbb{F}_\theta(x)$ for all $x \in X$, and likewise, $\mathbb{F}'_\theta(X)$ be the intersection of $\mathbb{F}'_\theta(x)$.*

The forgetting set indicates the set of parameter perturbations that lead to a classifier that misclassifies the sample $x$. During training, we aim to prevent parameter updates $\Delta\theta$ from falling within the forgetting set. Thus, we can formulate the problem of learning a new task as follows:

**Definition 2** (Certifiable Continual Learning). *Let $X, Y \sim \mathcal{T}_k$ be a set of samples from the new task, and $\tilde{X}, \tilde{Y} \sim \mathcal{T}_{0:k-1}$ be a set of samples from previous tasks, on which the classifier $f_\theta$ is trained. We wish to solve the following optimization problem*

$$\min_{\Delta\theta} \mathcal{L}(f_{\theta+\Delta\theta}(X), Y) \qquad\qquad s.t. \ \Delta\theta \in \mathbb{F}'_\theta(\tilde{X}) \qquad\qquad (1)$$

Intuitively, we seek to minimize the loss on the samples of the new task, while maintaining that the previous samples are not forgotten. Heuristic approaches, such as ER (Chaudhry et al., 2019), attempt to achieve this by simply including $\tilde{X}$ within the training loss, but offer no guarantees that a previously learned sample will not be forgotten. In this work, we are interested in providing theoretical guarantees; hence, the above definitions serve as a general framework to theoretically ground the problem of certifiable continual learning. Next, we propose a concrete method using LiRPA to perform certifiable continual learning.

# 4 CERTIFIABLE CONTINUAL LEARNING (CERCE)

In this section, we propose **Cer**tifiable **C**ontinual **LE**arning (CerCE), a novel method to perform continual learning, while providing certificates against forgetting during training. The key idea is to **reinterpret model parameter updates as perturbations to the weights**. With that in mind, let us present a special case of the auto-LiRPA (Xu et al., 2020) theorem for perturbing model weights:

**Theorem 1.** *Given fixed input batch $X \in \mathbb{R}^{n \times d}$, a model parameterized by $\theta = \{\theta^{(1)}, ..., \theta^{(\omega)}\}$, and a perturbation radius set $\gamma = \{\gamma^{(1)}, ..., \gamma^{(\omega)}\}$ where $\gamma^{(i)} \in \mathbb{R}$, and a function of the model outputs $h(f_\theta(X))$, we can obtain LiRPA coefficients $\underline{W}, \underline{b}, \overline{W}, \overline{b}$ such that*

$$\underline{b} + \underline{W}(\theta + \Delta\theta) \leq h(f_{\theta+\Delta\theta}(X)) \leq \overline{b} + \overline{W}(\theta + \Delta\theta)$$

*if $\|\Delta\theta\|_p \leq \gamma := \forall i : \|\Delta\theta^{(i)}\|_p \leq \gamma^{(i)}$ .*

Note that $\underline{W} \in \mathbb{R}^{n \times (c-1) \times K}, \underline{b} \in \mathbb{R}^{n \times (c-1)}, \overline{W} \in \mathbb{R}^{n \times (c-1) \times K}, \overline{b} \in \mathbb{R}^{n \times (c-1)}$ are functions of $\theta, X$, and $\gamma$. However, for the sake of brevity, we omit this from our notation throughout the paper. We arrive at this special case simply by substituting bounded norm perturbations to the network weights in the LiRPA framework. A simple proof is presented in Section B. The theorem states that within the given perturbation set, the function of the model output $h(f_\theta(x))$ is bounded by two linear functions of $\theta$. We can use these linear bounds to achieve our goal of preventing forgetting. Our main result is the corollary below, which allows for performing weight updates on a neural network while certifying that it will not forget samples for which the bound is satisfied.

**Corollary 1.1.** *Given input batch $\tilde{X} \in \mathbb{R}^{n \times d}$, labels $\tilde{Y} \in \mathbb{R}^n$, weight update $\Delta\theta$, and lower-bound coefficients from LiRPA $\underline{W}, \underline{b}$, corresponding to $h(f_\theta(\tilde{X}))$, and perturbation set $\gamma$, then $\Delta\theta \in \mathbb{F}'_\theta(\tilde{X})$ if $\|\Delta\theta\|_p \leq \gamma$ and $0 \leq \underline{b} + \underline{W}(\theta + \Delta\theta)$. $n, d, c$ are the batch size, input dimension and number of classes respectively, and $h(f_\theta(\tilde{X})) \in \mathbb{R}^{n \times (c-1)}$ is defined as follows*

$$h(f_\theta(\tilde{X}))_{i,j} = \begin{cases} f_\theta(\tilde{X}_i)_{\tilde{Y}_i} - f_\theta(\tilde{X}_i)_j, & \text{if } j < \tilde{Y}_i, \\ f_\theta(\tilde{X}_i)_{\tilde{Y}_i} - f_\theta(\tilde{X}_i)_{j+1}, & \text{if } j \geq \tilde{Y}_i \end{cases}$$

The function $h$ above simply subtracts the prediction scores of each class from that of the true class; i.e., the difference between the prediction score of the true class and any other class. Thus, if the lower bound is satisfied, that means the true class has the highest prediction score, and the sample is being classified correctly.

Corollary 1.1 suggests that as long as the lower bound resulting from LiRPA is non-negative, and we constrain the magnitude of the weight update to the model (e.g., by gradient clipping), the model will not misclassify the sample set $\tilde{X}$. This means we may take arbitrary gradient steps that include additional terms (e.g., weight decay, momentum, auxiliary loss terms) so long as the constraints are satisfied. Following Corollary 1.1, we can restate the constraints for the optimization problem in Eq. (1) as follows:

$$\min_{\Delta\theta} \ \mathcal{L}(f_{\theta+\Delta\theta}(X), Y) \qquad\qquad s.t. \ 0 \leq \underline{b} + \underline{W}(\theta + \Delta\theta), \ \|\Delta\theta\|_p \leq \gamma \qquad\qquad (2)$$

where $\underline{W}, \underline{b}$ are the LiRPA coefficients of the lower bound corresponding to $h(f_\theta(\tilde{X}))$ and $\theta$ with perturbation $\gamma$ and $X, Y$ are samples from the current task (See Fig. 1). The solutions to the above optimization problem are a subset of Eq. (1), and it leaves us with linear constraints that are easier to handle. However, since the loss function is still not linear, solving this optimization problem is non-trivial. Hence, we detail our optimization strategy in the following section.

## 4.1 OPTIMIZATION STRATEGY

**Gradient Distance Minimization** Neural networks are typically trained using non-linear loss functions via stochastic gradient descent. As such, let $X, Y$ be the batch of current task input samples and $\nabla_\theta \mathcal{L}(f_\theta(X), Y)$ be the gradient with respect to the training loss function. We can reformulate the optimization problem in Eq. (2) as an optimization problem for each step of the gradient weight update. We propose finding the update direction with the minimum distance from the gradient $\nabla_\theta \mathcal{L}(f_\theta(X), Y)$ while satisfying the constraints

$$\min_{\Delta\theta} \left\| \Delta\theta - \nabla_\theta \mathcal{L}(f_\theta(X), Y) \right\|_q \qquad s.t.\ 0 \le \underline{b} + \underline{W}(\theta + \Delta\theta),\ \|\Delta\theta\|_p \le \gamma \qquad (3)$$

where $q \in \{1, 2, \infty\}$ is the type of norm (e.g., $q = 2$ corresponds to a projection of the gradient onto the feasible set, note that $p$ and $q$ are arbitrary norms and not dependent). Depending on the choice of $q$, Eq. (3) has a linear/quadratic objective with linear/norm constraints and can be solved efficiently.

**Lagrangian** For the case of large neural networks with millions of parameters, constrained optimization can be infeasible or very costly. Thus, instead of Eq. (3), we propose to optimize the Lagrangian using unconstrained stochastic gradient descent. Doing so means the only additional computational burden is that of computing the LiRPA coefficients $\underline{W}, \underline{b}$. That leaves us with the following objective:

$$\mathcal{L}_{CerCE} = \mathcal{L}(f_\theta(X), Y) + \lambda \sum_{i,j} \max(0, -(\underline{W}\theta + \underline{b}))_{i,j} \qquad (4)$$

The first term is simply the loss function on the current task data, and the second term is optimizing the unsatisfied constraints with a scaling hyperparameter $\lambda$, where $\max(\cdot, \cdot)$ is the element-wise maximum, and the sum is taken over the dimensions of the resulting tensor where $i \le n$, $j \le c - 1$. While the above objective does not guarantee that the constraints will be satisfied, our certificates of Corollary 1.1 still hold for any constraint that is satisfied during training. We show that this objective works well in practice in the experiment section.

## 4.2 USE OF BUFFER

Equation (2) requires the LiRPA coefficients to be computed with respect to past task examples $\tilde{X}$. There are two possible approaches to achieve this, considering that in CL, we only have access to data from the current task, not previous ones. The first approach is to compute the coefficients at the end of the training phase of each task and store only the coefficients and use them for the rest of the training. This approach would be ideal if not for one major flaw: since the linear relaxation depends on the current parameters $\theta$, the perturbation radius $\gamma$ would need to be large enough to cover the entire parameter space used for all upcoming tasks. This would lead to extremely loose lower bounds, which are impractical since the corresponding constraints will never be satisfied (i.e., in a large radius around the parameters, there are likely to be points that misclassify the inputs). Thus, we will take the second approach, a common practice in CL literature: keeping a small buffer of samples from previous tasks to be used during training new tasks, namely, rehearsal-based approaches.

However, this approach comes with its challenges. Firstly, we must choose which samples to store in the buffer. A common practice is to use reservoir sampling (Aggarwal, 2006) to keep the distribution of the buffer the same as the past data distribution. In Section 5.2, we experiment with additional filters for sample selection and find that simple reservoir sampling works best in practice. A second challenge is *memory overfitting*, where the network overfits to the samples stored in memory and fails to generalize well. In the following section, we will provide theoretical justifications that our method may lead to reduced memory overfitting compared to typical rehearsal-based methods.

## 4.3 TIGHTER PAC-BAYES GENERALIZATION BOUNDS

The PAC-Bayes generalization bound mentioned in Theorem 4 (Section C) provides an upper bound on the generalization error of a learning algorithm. Using this formulation, and setting the training

data, $\mathcal{S}$, to be the memory buffer, and assuming that the buffer follows the same distribution as the joint distribution of all tasks (which we can easily achieve using reservoir sampling), the PAC-Bayes bound will hold. Below, we propose a modified version of the PAC-Bayes bound that suggests the improved generalization of CerCE:

**Theorem 2.** *Let $\theta_0$ be the parameter initialization and $\theta^*$ be the final parameter vector after training. Then the following bound holds:*

$$E_{\theta \sim \mathcal{N}(\theta^*, \sigma_\mathcal{Q} I)}\big[l_{\mathcal{T}_{0:k}}(\theta)\big] \leq e^{-C_1} \cdot \max_{\|\Delta\theta\|_2 \leq \gamma} \hat{l}_\mathcal{M}(\theta^* + \Delta\theta) + e^{-C_2} \tag{5}$$

$$+ \sqrt{\frac{KL(\mathcal{N}(\theta^*, \sigma_\mathcal{Q} I) || \mathcal{N}(\theta_0, \sigma_\mathcal{P} I)) + \log \frac{2\sqrt{\tilde{n}}}{\delta}}{2\tilde{n}}}$$

*with probability $1 - \delta$ over the draw of buffer samples, where $\mathcal{M}$ is the set of buffer samples sampled from previous tasks $\mathcal{T}_i$, $i \in \{0, ..., k\}$, and $\tilde{n} = |\mathcal{M}|$ is the number of buffer samples, and*

$$C_1 = \frac{(m - \frac{\gamma^2}{\sigma_\mathcal{Q}^2})^2}{4m}, \ C_2 = \frac{-2\sqrt{m} + \sqrt{-4m + \frac{8\gamma^2}{\sigma_\mathcal{Q}^2}}}{4}$$

*assuming $\frac{\gamma}{\sigma_\mathcal{Q}} > m$, with $m$ being the number of parameters in $\theta$, and $\sigma_\mathcal{Q}$, $\sigma_\mathcal{P}$ the hyper-parameters of the gaussian distributions.*

The above theorem is the result of substituting standard Gaussians into the original PAC-Bayes bound and partitioning the resulting parameter distribution into inside and outside the perturbation radius $\gamma$. See Section C for more details and proof. Immediately, it follows that if the LiRPA constraints are satisfied, the above bound is tighter, since the first term on the right-hand side disappears completely.

**Corollary 2.1** (Tighter PAC-Bayes Bound)**.** *Given LiRPA coefficients $\underline{W}, \underline{b}$ corresponding to $h(f_\theta(\mathcal{M}))$, and perturbation radius $\gamma$, if $0 \leq \underline{W}\theta^* + \underline{b}$ then with probability $1 - \delta$*

$$E_{\theta \sim \mathcal{N}(\theta^*, \sigma_\mathcal{Q} I)}\big[l_{\mathcal{T}_{0:k}}(\theta)\big] \leq e^{-C_2} + \sqrt{\frac{KL(\mathcal{N}(\theta^*, \sigma_\mathcal{Q} I) || \mathcal{N}(\theta_0, \sigma_\mathcal{P} I)) + \log \frac{2\sqrt{\tilde{n}}}{\delta}}{2\tilde{n}}} \tag{6}$$

This is due to the linear lower bound being valid in the entire epsilon ball with radius $\gamma$ centered at $\theta^*$. This means that optimizing to satisfy the LiRPA lower bounds leads to a tighter generalization bound, suggesting less overfitting on the memory buffer, which is further validated by our improved performance compared to ER (Chaudhry et al., 2019) in the experiment section.

## 4.4 Additional Implementation Details

**Slack Variables** In Eq. (3), it could be the case that not all the constraints for every sample can be satisfied simultaneously. To guarantee a feasible solution, we can add a slack term $\zeta \in \mathbb{R}^n$ to the constraints and minimize the norm of the slack variables,

$$\min_{\Delta\theta, \zeta} \ \big\|\Delta\theta - \nabla_\theta \mathcal{L}(f_\theta(X), Y)\big\|_q + c \cdot \big\|\zeta\big\|_{q'} \qquad s.t. \ 0 \leq \underline{b} + \underline{W}(\theta + \Delta\theta) + \zeta \tag{7}$$

$$\|\Delta\theta\|_p \leq \gamma, \ 0 \leq \zeta$$

where $c$ is a scaling constant. This does not significantly affect the computational complexity of the optimization problem and guarantees a feasible solution at the cost of potentially violating some of the certificates.

**Inclusion of Buffer Samples** Our optimization strategy does not make any strong assumptions on the loss function, and so we find that including current task samples in the training loss, i.e., $\mathcal{L}$, can help with training without loss of generality. We investigate the effect of this choice in an ablation study in Section 5.2. We find that CerCE performs only marginally worse without including buffer samples in the cross-entropy loss, demonstrating that this choice does not undermine the soundness and effectiveness of our original methodology. We provide a pseudocode of CerCE in Algorithm 1. We include our code in the supplementary material and will make it publicly available upon acceptance.

---

**Algorithm 1** CerCE: Certifiable Continual Learning via LiRPA Constraints

---

1: **Initialize:** model parameters $\theta$, perturbation radius $\gamma$, replay buffer $\mathcal{B} \leftarrow \emptyset$, learning rate $\alpha$
2: **for** each task $k = 0, 1, 2, \ldots$ **do**
3:     **for** each minibatch $(X, Y)$ from current task $\mathcal{T}_k$ **do**
4:         $\tilde{X}, \tilde{Y} \leftarrow \mathcal{B}$                                       ▷ Sample buffer minibatch
5:         $\mathcal{L}_{\text{ce}} \leftarrow \mathcal{L}(f_\theta([X, \tilde{X}]), [Y, \tilde{Y}])$                ▷ Cross Entropy loss
6:         $\underline{W}, \underline{b} \leftarrow LiRPA(f, \theta, \tilde{X}, \tilde{Y}, \gamma)$        ▷ Compute LiRPA coefficients
7:         **if** using constrained optimization **then**
8:             $\Delta\theta \leftarrow$ Solve Eq. (7)
9:             $\theta \leftarrow \theta - \alpha\Delta\theta$                       ▷ Update $\theta$ using $\Delta\theta$
10:        **else**                               ▷ Lagrangian relaxation
11:             $\mathcal{L}_{\text{CerCE}} \leftarrow \mathcal{L}_{\text{ce}} + \lambda \sum_{i,j} \max(0, -(\underline{W}\theta + \underline{b}))_{i,j}$
12:             Update $\theta$ via gradient descent using $\nabla_\theta \mathcal{L}_{\text{CerCE}}$
13:         **end if**
14:         Update replay buffer $\mathcal{B}$ with new samples $(X, Y)$ (e.g., reservoir sampling)
15:     **end for**
16: **end for**

---

## 5 EXPERIMENTS

**Baselines** As CerCE is one of the first to provide a rigorous framework for continual learning certificates, there are few competitors to our method. InterContiNet (Wołczyk et al., 2022) is one such method, although they do not make use of a buffer. Additionally, their method certifies that the worst-case performance of their network does not get worse; however, performance is evaluated in weight interval centers, which leaves room for further fluctuations. As for non-certified methods, we also compare to EWC (Kirkpatrick et al., 2017) and LwF (Li & Hoiem, 2017) as classical buffer-less baselines, as well as ER (Chaudhry et al., 2019), DER++ (Buzzega et al., 2020), and LPR (Yoo et al., 2024) as representatives of rehearsal-based baselines. We additionally compare to A-GEM (Chaudhry et al., 2018), which uses gradient projections on a memory buffer. As the focus of our work is to lay the groundwork for certifiable continual learning, the purpose of these comparisons is to demonstrate the effectiveness of our method compared to heuristic baselines while offering certificates, not to claim state-of-the-art performance. Finally, we include the "joint" baseline, training with access to the data from all tasks simultaneously, as an upper bound on performance, in addition to the "naive" baseline, which is training sequentially without any CL methods, as a lower bound for comparison.

**Metrics** We use two different metrics for measuring performance. First, the standard Final Average Accuracy (FA), the average accuracy of the final model across all tasks after training on the last task is completed. Second, we propose a new metric specific to measuring the certification performance of each method: Average Certification (AC): defined as the average ratio of samples in the buffer that satisfy the certification constraints at the end of each epoch during training. Note that even though most baselines do not provide certificates, we can still measure our certificates for these baselines by keeping a buffer and computing the LiRPA constraints without including them in the training objective. For the exact definition of metrics, refer to Section D.

**Network Architectures** Our methodology presented in the previous section does not assume any particular constraints for architecture or training details beyond being compatible with LiRPA frameworks. However, while currently auto-LiRPA (Xu et al., 2020) supports convolution and self-attention, it does not support weight perturbation for these operations. Given the sophistication of the LiRPA framework (as briefly discussed in Section G), we believe the implementation of such methods to be beyond the scope of this work. As support for these operations is added in the future, CerCE will be applicable to the corresponding architectures without further modification. However, to perform experiments on standard CL datasets, we make use of frozen pretrained encoders and train an MLP on top as the classifier. For image datasets, we use a Vision Transformer (ViT) (Dosovitskiy et al., 2021), and for text, we use SentenceBERT (Reimers & Gurevych, 2019). We want to emphasize that our methodology and formulations are not constrained to MLPs, and upon implementations of weight perturbation LiRPA for convolutional and attention-based operators, CerCE can be directly applied to a wide variety of architectures without requiring any further formulation.

**Datasets** While the current limitations of weight perturbation LiRPA prevent us from experimenting on large datasets, we conduct standardized experiments on a variety of image and text datasets. For benchmarking MLPs without a pretrained encoder, we include MNIST (Deng, 2012) and Fashion-MNIST (Xiao et al., 2017). For more standard image datasets, and by using a pretrained encoder, we experiment on CIFAR10, CIFAR100 (Krizhevsky et al., 2009), and TinyImagenet (Deng et al., 2009). Additionally, as certifiable machine learning is crucial to safety-critical environments, we experiment on a set of real-world safety-critical datasets. In line with previous works on neural network verification (Casadio et al., 2024), we use RUARobot (Gros et al., 2021), a set of user queries where the task is to determine whether or not a dialogue agent needs to disclose that it is not a human, as may be required by laws and regulations, as well as Medical (Abercrombie & Rieser, 2022) where the task is to identify whether or not the query of a user is indicative of a serious medical emergency, in which case taking immediate action to inform emergency services may be necessary. For all datasets and experiments, we follow the challenging class-incremental CL scenario, where new classes are added each task, and task labels are not provided during inference. For more experimental details and hyperparameters, refer to Section D.

## 5.1 MAIN RESULTS

**Image Datasets** We conduct experiments on standard image classification datasets. As can be seen in Table 1, CerCE outperforms InterContiNet (Wołczyk et al., 2022), the only other baseline with guarantees, due to compatibility with a buffer and higher flexibility, and is competitive with state-of-the-art rehearsal-based baselines while offering high certification rates, while existing baselines do not. Overall, CerCE achieves the best trade-off between accuracy and certification.

Table 1: Final Accuracy and Average Certification on Image datasets, all using 500 buffer samples. CerCE provides competitive accuracy while yielding significantly higher certification ratios throughout training. Bold indicates the best result, and the runner-up is underlined. * Certificate is on the worst-case performance, but during evaluation, the worst-case is not measured.

| Method | Buffer | Cert. | MNIST FA (↑) | MNIST AC (↑) | FashionMNIST FA (↑) | FashionMNIST AC (↑) | CIFAR10 FA (↑) | CIFAR10 AC (↑) | CIFAR100 FA (↑) | CIFAR100 AC (↑) | TinyImagenet FA (↑) | TinyImagenet AC (↑) |
|---|---|---|---|---|---|---|---|---|---|---|---|---|
| Joint | - | - | 97.71 ±0.17 | - | 85.56 ±0.66 | - | 74.09 ±3.64 | - | 57.55 ±0.40 | - | 82.08 ±0.80 | - |
| Naive | - | - | 19.03 ±0.04 | 11.34 ±0.08 | 19.91 ±0.00 | 13.44 ±0.22 | 18.61 ±0.53 | 8.89 ±1.10 | 8.84 ±0.04 | 2 ±0.22 | 10.64 ±0.00 | 0.40 ±0.09 |
| EWC | ✗ | ✗ | 19.05 ±0.02 | 11.54 ±0.53 | 19.91 ±0.00 | 14.04 ±1.03 | 18.70 ±0.58 | 8.48 ±0.01 | 8.82 ±0.06 | 1.89 ±0.07 | 11.13 ±0.59 | 1.45 ±0.02 |
| LwF | ✗ | ✗ | 19.09 ±0.00 | 12.42 ±0.11 | 19.90 ±0.01 | 10.96 ±0.70 | 18.73 ±0.30 | 9.52 ±0.01 | 8.77 ±0.08 | 3.61 ±0.63 | 9.41 ±0.00 | 1.88 ±0.16 |
| InterContiNet | ✗ | ✓ | 40.73 ±3.26 | 100* | 35.11 ±0.02 | 100* | 19.07 ±0.15 | 100* | 9.42 ±0.01 | 100* | 9.24 ±0.23 | 100* |
| AGEM | ✓ | ✗ | 28.49 ±1.18 | 6.84 ±1.01 | 32.08 ±1.70 | 19.94 ±0.43 | 38.49 ±0.84 | 14.68 ±0.22 | 10.26 ±1.72 | 1.34 ±0.07 | 19.39 ±0.52 | 1.06 ±0.15 |
| ER | ✓ | ✗ | 85.26 ±1.49 | 1.12 ±0.52 | 76.43 ±1.02 | 0.7 ±0.40 | 52.66 ±0.80 | 21.67 ±2.11 | 22.9 ±0.38 | 1.46 ±0.69 | 55.49 ±1.30 | 1.06 ±0.08 |
| DER++ | ✓ | ✗ | 85.66 ±1.09 | 0.00 ±0.00 | **77.75** ±1.18 | 0.00 ±0.00 | 53.14 ±1.52 | 4.91 ±1.25 | **24.50** ±0.66 | 0.00 ±0.00 | **61.21** ±0.85 | 0.18 ±0.01 |
| LPR | ✓ | ✗ | 85.08 ±1.58 | 0.80 ±0.98 | 75.77 ±1.93 | 0.30 ±0.26 | 52.83 ±1.58 | 19.10 ±0.15 | 22.63 ±0.16 | 1.22 ±0.59 | 50.15 ±0.81 | 0.00 ±0.00 |
| CerCE | ✓ | ✓ | **86.57** ±0.94 | **90.5** ±2.28 | 76.05 ±0.94 | **91.56** ±1.73 | **54.45** ±0.77 | **91.17** ±0.29 | 23.49 ±0.43 | **94.55** ±0.23 | 53.18 ±0.33 | **98.58** ±0.17 |

**Safety-critical Text Data** In order to demonstrate the effectiveness of our method in real-world safety-critical scenarios, we conduct experiments on two text classification datasets. In line with previous works (Casadio et al., 2024), we use a pre-trained text-encoder, Sentence-BERT (Reimers & Gurevych, 2019), to transform sentences to vector embeddings and train an MLP as the classifier. The results are presented in Table 2 and demonstrate the effectiveness of CerCE in continual learning for real-world safety-critical scenarios. Detailed descriptions of the datasets can be found in Section F.

Table 2: Safety-critical Datasets. CerCE provides significantly higher certification rates while maintaining accuracy. Bold indicates the best result, and the runner-up is underlined. * Certificate is on the worst-case performance, but during evaluation, the worst-case is not measured.

| Method | RUARobot FA (↑) | RUARobot AC (↑) | Medical FA (↑) | Medical AC (↑) |
|---|---|---|---|---|
| Joint | 99.39 ±0.00 | - | 99.36 ±0.00 | - |
| Naive | 50 ±0.29 | 30.20 ±0.98 | 49.14 ±0.7 | 28.63 ±1.60 |
| EWC | 50 ±0.00 | 48.42 ±0.11 | 54.97 ±10.53 | 28.63 ±2.09 |
| LwF | 50 ±0.00 | 28.48 ±0.01 | 50.34 ±0.00 | 37.12 ±0.27 |
| InterContiNet | 50 ±0.12 | 100* | 50.85 ±0.45 | 100* |
| AGEM | 56.85 ±1.52 | 39.78 ±1.29 | 86.59 ±1.58 | 68.79 ±1.43 |
| ER | **89.08** ±1.45 | 37.02 ±3.24 | 98.37 ±1.22 | 39.47 ±2.33 |
| DER++ | 85.69 ±0.16 | 22.32 ±0.66 | 98.69 ±0.54 | 58.62 ±1.22 |
| LPR | 88.73 ±1.69 | 33.67 ±0.55 | 97.58 ±0.32 | 32.18 ±3.43 |
| CerCE | 89.03 ±0.42 | **79.84** ±1.66 | **98.83** ±0.61 | **82.94** ±2.93 |

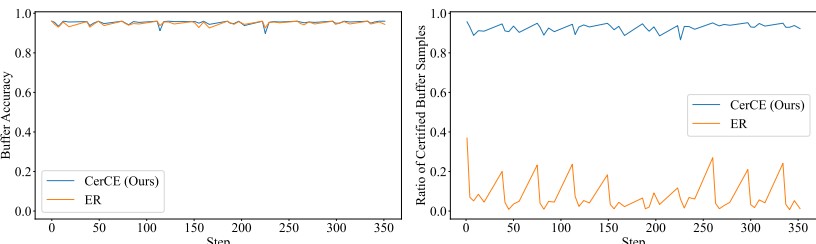

Figure 2: Accuracy and ratio of certified samples in the Buffer for CerCE and ER over time. Not only does CerCE result in high buffer accuracy over time, it results in a significant portion of the buffer being certified as opposed to ER.

## 5.2 ABLATION STUDIES AND EXPLORATORY EXPERIMENTS

**Ratio of Certified Samples in the Buffer** In order to empirically measure the success of CerCE in preventing forgetting and providing certificates during training, we plot the ratio of correctly classified as well as certified samples within the input buffer through time on the CIFAR10 dataset in Fig. 2. We also include the certification rate of ER as a comparison. While ER does not train and provide certificates, it is possible to use our method to simply check whether or not the correct classification of a sample is certified during training. As seen in Fig. 2, while both methods result in a high buffer accuracy, CerCE yields a significantly higher portion of samples in the buffer that are certified, which validates our main contribution: certified prevention of forgetting during continual learning.

**Per-class Certification Rate and Backward Generalization** To further investigate the empirical connection of our certificates with existing metrics, we investigate the relationship between the ratio of samples that are certified within the buffer for each class and the classification accuracy on that class across consecutive tasks. We expect to see a positive correlation between the ratio of certified samples and the test set accuracy for each class. We plot the results for the TinyImagenet dataset, which consists of 200 classes, and use CerCE for training. For this experiment, we used 5000 buffer samples to make sure each of the 200 classes is represented sufficiently in the buffer; no other hyperparameters were changed. The results are presented in Fig. 3, and a clear correlation is observed between buffer certification ratio and accuracy across all tasks, with a positive Pearson correlation coefficient (Pearson & Galton, 1895). We conduct a similar experiment and discussion while using ER as the training method in Section D, and to summarize, the results are consistent with our observations in Fig. 3.

**Buffer and Loss Function Sample Selection** As our certificates are provided on the samples from the buffer, it is natural to investigate the effect of different sampling schemes on selecting memory samples. Thus, we propose three different filtering schemes to be applied on top of reservoir sampling (which is applied in all scenarios): No additional filter (random), only include samples that the model classifies correctly (correct), and only include the samples for which the LiRPA constraints are satisfied (bound). In addition, we experiment with whether or not to include the buffer samples in the cross-entropy loss alongside the current task samples. The results are presented in Table 3. First, we see that CerCE is not very sensitive to the inclusion of new samples in the cross-entropy loss, validating the effectiveness of our constraint-based loss term. Additionally, we see that selecting correct or bound samples can have a marginal positive impact on certification, while incurring additional computational cost. We hypothesize that this is because it is easier to certify already certified/correctly classified samples rather than random ones.

For additional ablation studies, exploratory experiments, and details on the hyperparameters, refer to Section D.

## 6 CONCLUSION

In this work, we introduced Certifiable Continual Learning (CerCE), a novel framework that brings formal guarantees to the problem of continual learning. By leveraging Linear Relaxation Perturbation

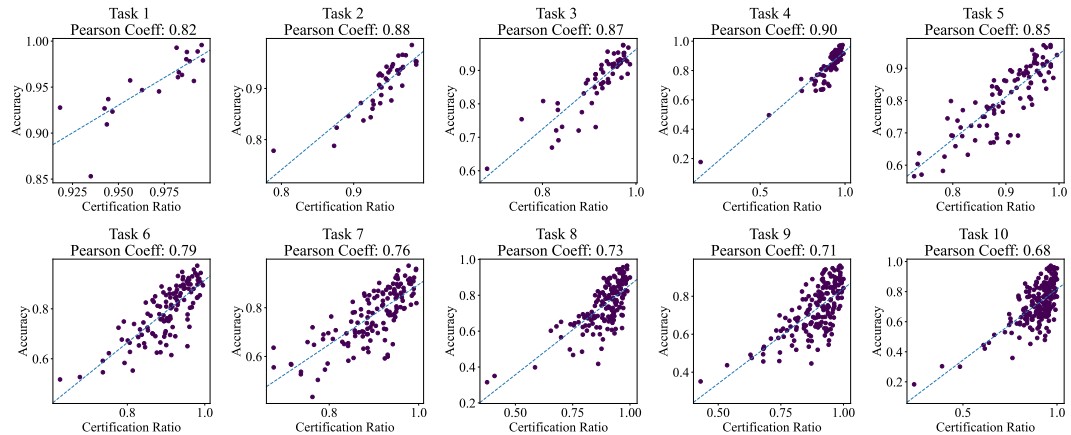

Figure 3: Correlation of per-class certification ratio in the buffer and test set accuracy after the training of each task on the TinyImagenet dataset training using CerCE with 5000 samples. Each point represents one class, the dashed line represents the optimal linear regressor, and the Pearson Correlation Coefficient is denoted for each plot. Each plot shows the per-class correlation after training was finished on each task, for all seen classes thus far.

Table 3: Ablation Study of what is included in the buffer, as well as what filter is applied to buffer sampling: random (no filter), correct (only correctly classified samples), bound (only samples which satisfy the certification bounds)

| Selection Filter | Random | | Correct | | Bound | |
|---|---|---|---|---|---|---|
| | FA(↑) | AC(↑) | FA(↑) | AC(↑) | FA(↑) | AC(↑) |
| Current Task Samples | 81.51 | 81.30 | 80.68 | 80.96 | 78.78 | 80.44 |
| All Samples | 86.57 | 90.5 | 85.94 | 90.87 | 83.94 | 90.86 |

Analysis (LiRPA) and formulating CL as a constrained optimization problem, CerCE enables learning new tasks without forgetting past knowledge, with verifiable certificates during training. Our method not only addresses catastrophic forgetting but also improves generalization by tightening PAC-Bayes bounds, helping mitigate memory overfitting, a persistent challenge in rehearsal-based methods. Through extensive experiments across standard benchmarks and real-world safety-critical datasets, we demonstrate that CerCE achieves competitive performance while uniquely offering provable non-forgetting guarantees. We believe this work lays essential groundwork for a new generation of continual learning algorithms where reliability, robustness, and certification are emphasized, a critical step for deploying CL systems in high-stakes applications.

As CerCE is the first to pursue certifiable continual learning through linear relaxations, there are limitations such as architectural compatibility, additional computational overhead, and reliance on a memory buffer. However, all of the above-mentioned limitations are not inherent to CerCE and can be improved with further development of faster and more compatible LiRPA algorithms that can calculate tighter upper and lower bounds, which are valid over larger radii in the parameter space. CerCE provides a necessary first step for future work to extend and improve certification in continual learning, which is necessary for deployment in real-world safety-critical systems.

## ETHICS STATEMENT

Safety-critical applications, such as healthcare and autonomous driving, often require guarantees on specific performance metrics to ensure people's safety. In this work, we take a step towards providing such guarantees for continual learning in terms of certifying non-forgetting of samples during training. However, it should be noted that while the certificates provided by CerCE are valid, they do not necessarily encompass all data, nor do they ensure the safety of the system by default, as it is up to the system designers to guarantee its safety. CerCE provides a framework for performing continual learning with theoretical guarantees and the assumptions, propositions, and limitations of the method

should be taken into account during any application. We hope that CerCE can serve as a stepping stone for designing safe and reliable AI systems with real-world applications.

## REPRODUCIBILITY STATEMENT

In this paper, we take thorough measures to ensure the reproducibility of our work. In addition to describing the method in detail in Section 4, we provide a pseudocode of our algorithm in Algorithm 1. We outline our experiment setup in Section 5, and provide additional details, including hyperparameters in Section D. Finally, we submit our code, including a README file for instructions, anonymously in the supplementary material and pledge to make it open-source on GitHub upon acceptance.

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

## A  TOY EXPERIMENT

To demonstrate the effectiveness of CerCE in preventing forgetting and provide some more intuition, we design a relatively simple toy example. We use a two-layer MLP with ReLU activations on binary classification of two-dimensional data. The tasks are split as demonstrated in Fig. 4, in two halves. A naive SGD approach leads to forgetting of the first task, while CerCE ensures that the samples kept within the buffer are correctly classified and the decision boundary of the network does not move past these samples (highlighted in red).

Additionally, we investigate which samples are more vulnerable to forgetting when not using CerCE. Figure 5 shows the samples that would be misclassified if we were to use a naive SGD approach for each epoch while training using CerCE. Naturally, samples closer to the decision boundary are more vulnerable.

## B  AUTO-LiRPA DETAILS

LiRPA is a framework for using linear relaxations in deriving certified bounds of neural network outputs under perturbations. Auto-LiRPA (Xu et al., 2020), provides a general framework for using these methods for general computational graphs and perturbations to the nodes of such graphs and the effects on the output. Below, we state deriving of LiRPA bounds as presented in Xu et al. (2020) as a theorem

**Theorem 3** (LiRPA (Xu et al., 2020)). *Let $V = \{v_i\}_{i=1}^k$ be a set of independent values, typically model inputs and parameters, such that they can take values from a perturbation set $\mathbb{S}_i$, i.e., $v_i \in \mathbb{S}_i$. For example, $\mathbb{S}_i = \{c_i\}$ if $v_i$ is a constant with no perturbation. Let $h_i(V)$ be a node in the computational graph, with $h_o(V)$ denoting the final output. Then, through LiRPA, we can obtain coefficients $\underline{W}_o, \overline{W}_o, \underline{b}_o, \overline{b}_o$ such that*

$$\underline{b}_o + \underline{W}_o V \le h_o(V) \le \overline{b}_o + \overline{W}_o V$$

*as long as $\forall i : v_i \in \mathbb{S}_i$.*

Below, we restate the special case of LiRPA (Theorem 1)

**Theorem 1.** *Given fixed input batch $X \in \mathbb{R}^{n \times d}$, a model parameterized by $\theta = \{\theta^{(1)}, ..., \theta^{(\omega)}\}$, and a perturbation radius set $\gamma = \{\gamma^{(1)}, ..., \gamma^{(\omega)}\}$ where $\gamma^{(i)} \in \mathbb{R}$, and a function of the model outputs $h(f_\theta(X))$, we can obtain LiRPA coefficients $\underline{W}, \underline{b}, \overline{W}, \overline{b}$ such that*

$$\underline{b} + \underline{W}(\theta + \Delta\theta) \le h(f_{\theta + \Delta\theta}(X)) \le \overline{b} + \overline{W}(\theta + \Delta\theta)$$

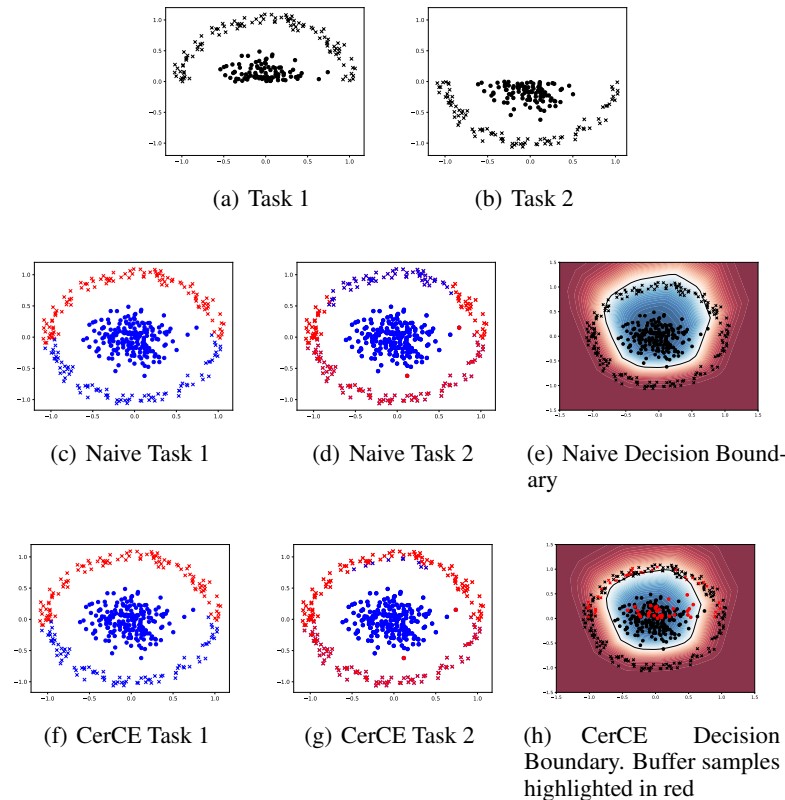

Figure 4: Non-linear Toy Example using a two layer MLP. A Naive approach forgets the first task, while CerCE maintains the decision boundary of the chosen buffer samples.

*if* $\|\Delta\theta\|_p \leq \gamma := \forall i : \|\Delta\theta^{(i)}\|_p \leq \gamma^{(i)}$ .

*Proof.* We can obtain this special case simply by setting $h_o(V) = h(f_\theta(X))$ and setting $\mathbb{S}$ for the $i$th parameter of the network to be the epsilon ball centered around $\theta^{(i)}$ with radius $\gamma^{(i)}$. Note that we do not assume any perturbations of the input $x$, that is, $\mathbb{S}_x = \{x\}$. $\qquad\square$

For the paper to be self-contained, we provide details of obtaining LiRPA coefficients as outlined in Xu et al. (2020), in Section G.

## C    PAC-BAYES DETAILS

**Theorem 4** (PAC-Bayes Generalization bound (McAllester, 1999))**.** *Let* $\mathcal{P}, \mathcal{Q}$ *be distributions over the hypothesis set (in our case, the **parameter space** of neural networks). Let* $\mathcal{S} \sim \mathcal{D}^n$ *be a set of* $n$ *samples drawn from the data distribution* $\mathcal{D}$. *Then, assuming* $\mathcal{P}$ *is independent of* $\mathcal{S}$, *we have:*

$$E_{\theta\sim\mathcal{Q}}\big[l_{\mathcal{D}}(\theta)\big] \leq E_{\theta\sim\mathcal{Q}}\big[\hat{l}_{\mathcal{S}}(\theta)\big] + \sqrt{\frac{KL(\mathcal{Q}||\mathcal{P}) + \log\frac{2\sqrt{n}}{\delta}}{2n}}$$

*with probability at least* $1 - \delta$ *over the draw of* $\mathcal{S}$.

The above theorem states, that the expected test error is bounded by above by two terms. First, the expected error on the training set $\mathcal{S}$, and second, how much the distribution $\mathcal{Q}$ deviates from $\mathcal{P}$. Often, $\mathcal{P}, \mathcal{Q}$ are referred to as *prior* and *posterior* in the literature. However, there is no constraint on $\mathcal{Q}$ to be the Bayesian posterior of $\mathcal{P}$ for the theorem to hold. We will show that our proposed method leads to a provable tightening of the first term. Following the above theorem, we can prove Theorem 2

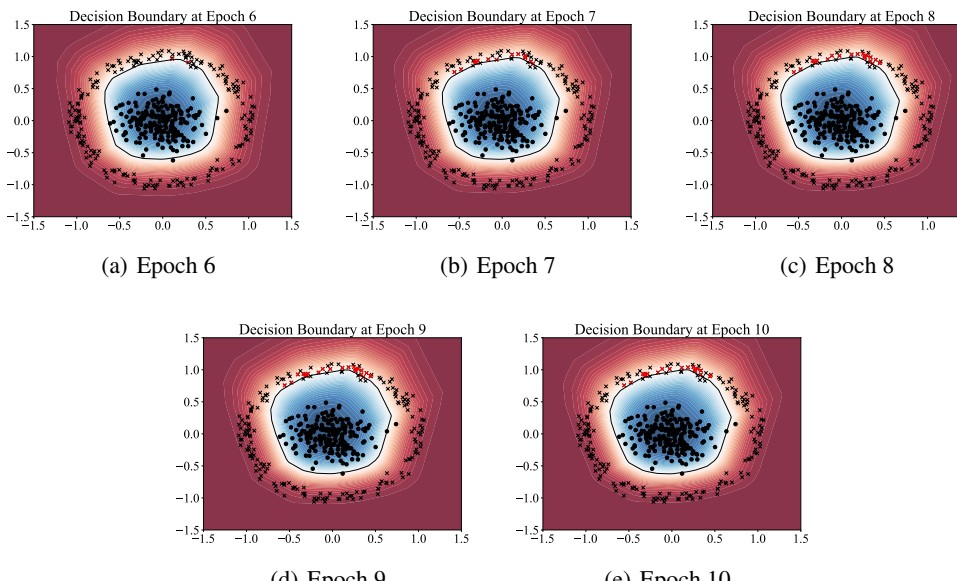

(a) Epoch 6        (b) Epoch 7        (c) Epoch 8

(d) Epoch 9        (e) Epoch 10

Figure 5: The last 5 epochs of the Non-linear Toy Example trained using CerCE. Highlighted samples in red are samples that would be misclassified if a naive approach were used for that epoch.

**Theorem 2.** *Let $\theta_0$ be the parameter initialization and $\theta^*$ be the final parameter vector after training. Then the following bound holds:*

$$E_{\theta \sim \mathcal{N}(\theta^*, \sigma_{\mathcal{Q}} I)}\big[l_{\mathcal{T}_{0:k}}(\theta)\big] \leq e^{-C_1} \cdot \max_{\|\Delta\theta\|_2 \leq \gamma} \hat{l}_{\mathcal{M}}(\theta^* + \Delta\theta) + e^{-C_2} \tag{5}$$

$$+ \sqrt{\frac{KL(\mathcal{N}(\theta^*, \sigma_{\mathcal{Q}} I) \| \mathcal{N}(\theta_0, \sigma_{\mathcal{P}} I)) + \log \frac{2\sqrt{\tilde{n}}}{\delta}}{2\tilde{n}}}$$

*with probability $1 - \delta$ over the draw of buffer samples, where $\mathcal{M}$ is the set of buffer samples sampled from previous tasks $\mathcal{T}_i$, $i \in \{0, ..., k\}$, and $\tilde{n} = |\mathcal{M}|$ is the number of buffer samples, and*

$$C_1 = \frac{(m - \frac{\gamma^2}{\sigma_{\mathcal{Q}}^2})^2}{4m}, \ C_2 = \frac{-2\sqrt{m} + \sqrt{-4m + \frac{8\gamma^2}{\sigma_{\mathcal{Q}}^2}}}{4}$$

*assuming $\frac{\gamma}{\sigma_{\mathcal{Q}}} > m$, with $m$ being the number of parameters in $\theta$, and $\sigma_{\mathcal{Q}}$, $\sigma_{\mathcal{P}}$ the hyper-parameters of the gaussian distributions.*

*Proof.* First consider Theorem 4, and let $\mathcal{S}$ be the memory buffer with $n$ samples, and $\mathcal{P}, \mathcal{Q}$ be $\mathcal{N}(\theta_0, \sigma_{\mathcal{P}} I), \mathcal{N}(\theta^*, \sigma_{\mathcal{Q}} I)$ respectively. Assuming the buffer uses reservoir sampling, the buffer distribution is the same as $\mathcal{T}_{0:k}$, hence the loss on the LHS is on $\mathcal{T}_{0:k}$, the distribution of all previous tasks. Here is the resulting intermediate form of Theorem 4 by way of these substitutions:

$$E_{\theta \sim \mathcal{N}(\theta^*, \sigma_{\mathcal{Q}} I)}\big[l_{\mathcal{T}_{0:k}}(\theta)\big] \leq E_{\theta \sim \mathcal{N}(\theta^*, \sigma_{\mathcal{Q}} I)}\big[\hat{l}_{\mathcal{M}}(\theta)\big] + \sqrt{\frac{KL(\mathcal{N}(\theta^*, \sigma_{\mathcal{Q}} I) \| \mathcal{N}(\theta_0, \sigma_{\mathcal{P}} I)) + \log \frac{2\sqrt{n}}{\delta}}{2n}}$$

Now we must show that the certificates being satisfied lead to a tighter bound. If $\Delta\theta = \theta - \theta^*$, since $\theta \sim \mathcal{N}(\theta^*, \sigma_{\mathcal{Q}} I)$, then $\|\Delta\theta\|_2^2$ follows a shifted and scaled chi-squared distribution, that is, $\|\Delta\theta\|_2^2 \sim \sigma_{\mathcal{Q}}^2 \chi_m^2$ where $m$ is the number of parameters in $\theta$. By law of total expectation we have:

$$E_{\theta \sim \mathcal{N}(\theta^*, \sigma_{\mathcal{Q}} I)}\big[\hat{l}_{\mathcal{M}}(\theta)\big] =$$

$$P(\|\Delta\theta\|_2 \geq \gamma) \cdot E_{\theta \sim \mathcal{N}(\theta^*, \sigma_{\mathcal{Q}} I)}\big[\hat{l}_{\mathcal{M}}(\theta) \big| \|\Delta\theta\|_2 \geq \gamma\big] +$$

$$P(\|\Delta\theta\|_2 \leq \gamma) \cdot E_{\theta \sim \mathcal{N}(\theta^*, \sigma_{\mathcal{Q}} I)}\big[\hat{l}_{\mathcal{M}}(\theta) \big| \|\Delta\theta\|_2 \leq \gamma\big]$$

By Lemma 1 of Laurent & Massart (2000) we have:

$$\forall t \in \mathbb{R}^+ : \ P(m - 2\sqrt{mt} \geq \frac{\|\Delta\theta\|^2}{\sigma_{\mathcal{Q}}^2}) \leq e^{-t}$$

$$\& \ \forall t' \in \mathbb{R}^+ : P(m + 2\sqrt{mt'} + 2t' \leq \frac{\|\Delta\theta\|^2}{\sigma_{\mathcal{Q}}^2}) \leq e^{-t'}$$

Now substitute the LHS of each inequality by $\frac{\gamma^2}{\sigma_{\mathcal{Q}}^2}$ and solve for $t, t'$:

$$m - 2\sqrt{mt} = \frac{\gamma^2}{\sigma_{\mathcal{Q}}^2} \Rightarrow t = \frac{(m - \frac{\gamma^2}{\sigma_{\mathcal{Q}}^2})^2}{4m}$$

Now for $t'$:

$$m + 2\sqrt{mt'} + 2t' = \frac{\gamma^2}{\sigma_{\mathcal{Q}}^2} \rightarrow 2t' + 2\sqrt{mt'} + m - \frac{\gamma^2}{\sigma_{\mathcal{Q}}^2} = 0$$

Solve quadratic equation:

$$t' = \frac{-2\sqrt{m} + \sqrt{-4m + \frac{8\gamma^2}{\sigma_{\mathcal{Q}}^2}}}{4}$$

Assuming that: $m < \frac{\gamma^2}{\sigma_{\mathcal{Q}}^2}$. Substitute in the intermediate bound:

$$E_{\theta \sim \mathcal{N}(\theta^*, \sigma_{\mathcal{Q}} I)}\big[\hat{l}_{\mathcal{M}}(\theta)\big] =$$
$$P(\|\Delta\theta\|_2 \geq \gamma) \cdot E_{\theta \sim \mathcal{N}(\theta^*, \sigma_{\mathcal{Q}} I)}\big[\hat{l}_{\mathcal{M}}(\theta)\big|\|\Delta\theta\|_2 \geq \gamma] +$$
$$P(\|\Delta\theta\|_2 \leq \gamma) \cdot E_{\theta \sim \mathcal{N}(\theta^*, \sigma_{\mathcal{Q}} I)}\big[\hat{l}_{\mathcal{M}}(\theta)\big|\|\Delta\theta\|_2 \leq \gamma]$$
$$\leq e^{-t} \cdot E_{\theta \sim \mathcal{N}(\theta^*, \sigma_{\mathcal{Q}} I)}\big[\hat{l}_{\mathcal{M}}(\theta)\big|\|\Delta\theta\|_2 \geq \gamma] +$$
$$e^{-t'} \cdot E_{\theta \sim \mathcal{N}(\theta^*, \sigma_{\mathcal{Q}} I)}\big[\hat{l}_{\mathcal{M}}(\theta)\big|\|\Delta\theta\|_2 \leq \gamma]$$

Now, since the first expectation is bounded by the maximum inside the radius, and the second is bounded by 1 (maximum of the error $l$) we have:

$$E_{\theta \sim \mathcal{N}(\theta^*, \sigma_{\mathcal{Q}} I)}\big[\hat{l}_{\mathcal{M}}(\theta)\big] \leq$$
$$e^{-t} \cdot \max_{\|\Delta\theta\|_2 \leq \gamma} \hat{l}_{\mathcal{M}}(\theta^* + \Delta\theta) +$$
$$e^{-t'} \cdot 1$$

$\square$

Corollary 1.1 follows immediately if bounds are satisfied, since we have that all samples must be classified correctly within $\|\Delta\theta\|_2 \leq \gamma$, then $\max_{\|\Delta\theta\|_2 \leq \gamma} \hat{l}_{\mathcal{M}}(\theta^* + \Delta\theta) = 0$.

$\square$

Note that $\sigma_{\mathcal{Q}}$ can be chosen freely, but $\sigma_{\mathcal{P}}$ must be chosen in a way such that $\mathcal{P}$ is independent of $\mathcal{S}$. A whole field of works is dedicated to deriving tighter and tighter PAC-Bayes bounds (Alquier et al., 2024), which is not the focus of our work. However, their contributions are largely applicable to our bound as well.

# D HYPERPARAMETERS, EXPERIMENTAL DETAILS, AND ADDITIONAL EXPERIMENTS

**Dataset splitting details** MNIST, FMNIST, and CIFAR10 were split into 5 tasks of 2 classes.

CIFAR100 was split into 10 tasks of 10, and TinyImagenet was split into 20 tasks of 10 classes. RUARobot and Medical both contain two classes and were split into two single-class tasks. All experiments were run on a single Nvidia A6000 GPU. **Hyperparameters** All experiments for CerCE were conducted using the lagrangian optimization variant, except for the toy example in Fig. 4, which used the constrained optimization technique. For the underlying LiRPA method, we used "crown+ibp" (Xu et al., 2020). Table 4 shows the hyperparameters used for each dataset. Buffer size was set to 500 samples for all methods using a buffer (including CerCE). For the experiments using pre-trained encoders, we used the ViT-B Dosovitskiy et al. (2021) architecture, and as for the MLP classifier, we used three layers with a hidden dimension of 400 and a ReLU activation in between.

Table 4: Hyperparameters for different datasets

| Dataset | MNIST | FMNIST | CIFAR10 | CIFAR100 | TinyImagenet |
|---------|-------|--------|---------|----------|--------------|
| lr | 0.1 | 0.1 | 0.01 | 0.01 | 0.01 |
| $\gamma$ | 0.002 | 0.002 | 0.001 | 0.001 | 0.0001 |
| $\lambda$ | 0.01 | 0.01 | 0.1 | 0.1 | 0.1 |
| epochs | 1 | 5 | 50 | 50 | 5 |

**Lambda & Gamma ablation** The results for various values of $\gamma$ and $\lambda$ are detailed in Table 5 and Table 6 respectively. For both $\gamma$ and $\lambda$, higher values consistently result in better performance metrics (AC as well as FA) until the value is too high for the objective to be feasible, and training does not converge to a desirable point.

Table 5: CerCE Average Certification (AC) and Final Average Accuracy (FA) on the MNIST dataset for different values of $\gamma$.

| $\gamma$ | 1e-5 | 1e-4 | 1e-3 | 0.01 |
|----------|------|------|------|------|
| AC ($\uparrow$) | 94.94 | 94.87 | 93.26 | 18.78 |
| FA ($\uparrow$) | 85.87 | 85.83 | 86.39 | 29.29 |

Table 6: CerCE Average Certification (AC) and Final Average Accuracy (FA) on the MNIST dataset for different values of $\lambda$.

| $\lambda$ | 0.0001 | 0.001 | 0.005 | 0.01 | 0.05 | 0.1 |
|-----------|--------|-------|-------|------|------|-----|
| AC ($\uparrow$) | 34.28 | 83.06 | 87.42 | 90.5 | 77.84 | 18.58 |
| FA ($\uparrow$) | 85.41 | 85.82 | 86.17 | 86.57 | 74.22 | 18.40 |

**Buffer Size Ablation** We experiment with varying buffer-sizes for CerCE and some existing methods on the MNIST dataset. The results are shown in Table 7. While certifying larger buffer sizes is more difficult as expected, CerCE still provides high certification rates as well as accuracy.

Table 7: CerCE Average Certification (AC) and Final Average Accuracy (FA) on the MNIST dataset for different buffer sizes.

| Buffer size | 100 | 500 | 1000 | 2000 |
|-------------|-----|-----|------|------|
| AC ($\uparrow$) | 95.60 | 90.5 | 89.92 | 88.03 |
| FA ($\uparrow$) | 70.02 | 86.57 | 89.14 | 90.92 |

**Per-class Correlation for ER**

We conduct the same experiment as Fig. 3 while using ER as the training scheme. We use the TinyImagenet dataset and 5000 samples in the buffer, keeping the experiment parameters consistent. The results are presented in Fig. 6. We observe that even though ER does not incorporate certificates into its training objective, and thus suffers from a low certification ratio, there is still a positive correlation between certification ratio and test set accuracy, further validating our intuition.

**Task Specific Accuracy Across Tasks**

For better contextualization of the accuracy and certification trade-off, and to complement Table 1, we plot the per-task test set accuracy across tasks for the MNIST and TinyImage datasets in Fig. 7 and

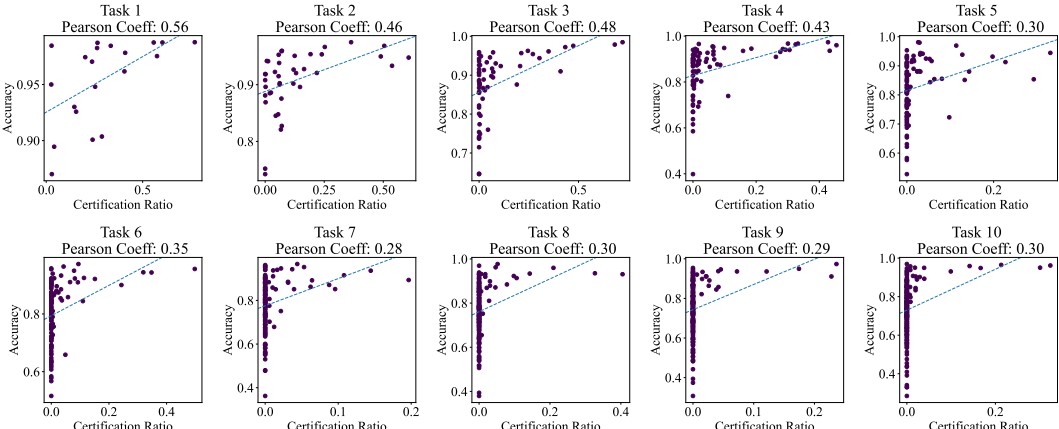

Figure 6: Correlation of per-class certification ratio in the buffer and test set accuracy after the training of each task on the TinyImagenet dataset training using ER with 5000 samples. Each point represents one class, the dashed line represents the optimal linear regressor, and the Pearson Correlation Coefficient is denoted for each plot. Each plot shows the per-class correlation after training was finished on each task, for all seen classes thus far.

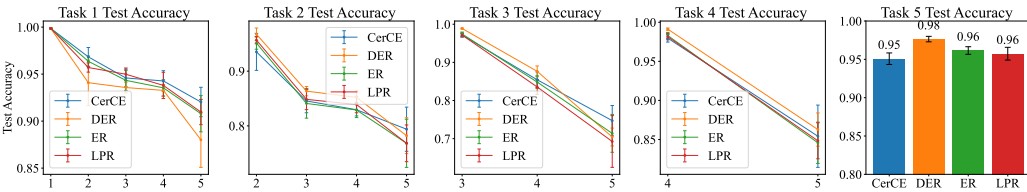

Figure 7: Test Set accuracies for the MNIST Dataset for various methods. Accuracy for each tasks test set is plotted across the training of each subsequent task.

Fig. 8 respectively. We observe that consistent with Table 1, CerCE is able to maintain competitive previous task accuracy to other buffer based methods, while providing certificates. Additionally, CerCE generally performs better on retaining past-task performance rather than improving newer task accuracies, which is in-line with our intuition since the focus of CerCE is providing formal certificates for past data samples.

**Metrics** Final Average Accuracy (FA) is defined below after training on $\tau$ tasks:

$$A_\tau = \frac{1}{k} \sum_{i=1}^{k} a_{i,\tau}$$

where $a_{i,\tau}$ is the accuracy of the test set of the $i$th task computed on the model after training on the $\tau$th task. Average Certification (AC) is defined below after $t$ epochs of training:

$$C_t = \frac{1}{t} \sum_{j=1}^{t} c_j$$

where $c_j$ is the ratio of buffer samples which are certified after $j$th training epoch ($0 < \underline{W}\theta_j + \underline{b}$ for that sample where $\theta_j$ is the parameters after epoch $j$). Note that $1...t$ includes all epochs from all tasks.

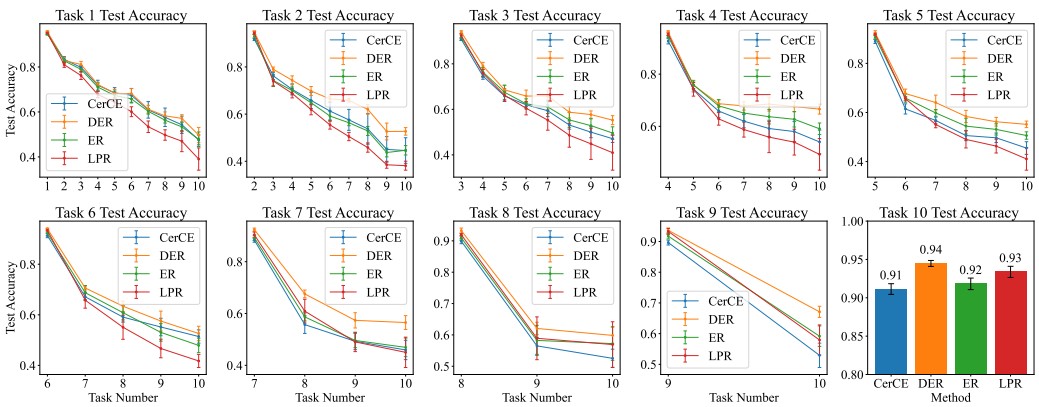

Figure 8: Test Set accuracies for the TinyImagenet Dataset for various methods. Accuracy for each tasks test set is plotted across the training of each subsequent task.

## E    RUNTIME COMPLEXITY

The majority of CerCE's computation time is due to computation of the LiRPA bounds. Computing LiRPA bounds consists of two stages: 1. propagating the perturbations forward through the network, which has O(r) complexity where O(r) is the complexity of a forward pass, and 2. Tightening the bounds through a backward pass and obtaining the coefficient, which, with the loss-fusion technique introduced in auto-LiRPA (Xu et al., 2020), also takes O(r) time. This, in general, leads to a 3-5 times slowdown (Xu et al., 2020) for computing the LiRPA coefficients. In our case, with the addition of the loss term, we get about 9x times slowdown compared to regular training, which is similar to that of InterContiNet. The training time for a single epoch of various datasets is detailed in Table 8.

Table 8: Running time for a single epoch in seconds for various methods on different datasets.

| Dataset | Naive | ER | InterContiNet | CerCE |
|---------|-------|------|---------------|-------|
| MNIST | 1.3 | 1.45 | 6.91 | 7.78 |
| CIFAR10 | 0.98 | 1.06 | 5.89 | 6.28 |
| CIFAR100 | 0.48 | 0.50 | 2.63 | 3.1 |
| TinyImageNet | 1.53 | 1.71 | 10.25 | 12 .03 |

## F    DATASET DETAILS

We follow the same setting as Casadio et al. (2024), they provide the following descriptions: **R-U-A-Robot (Gros et al., 2021)** The R-U-A-Robot dataset is a written English dataset consisting of 6800 variations on queries relating to the intent of 'Are you a robot?', such as 'I'm a man, what about you?'. The dataset was created via a context-free grammar template, crowd-sourcing and pre-existing data sources. It consists of 2,720 positive examples (where given the query, it is appropriate for the system to state its non-human identity), 3,400 negative/adversarial examples and 680 'ambiguous-if-clarify' examples (where it is unclear whether the system is required to state its identity). The dataset was created to promote transparency, which may be required when the user receives unsolicited phone calls from artificial systems. Given systems like Google Duplex, and the criticism it received for human-sounding outputs, it is also highly plausible for the user to be deceived regarding the outputs generated by other NLP-based systems. Thus we choose this dataset to understand how to enforce such disclosure requirements. We collapse the positive and ambiguous examples into one label, following the principle of 'better be safe than sorry', i.e., prioritising a high recall system. **Medical (Abercrombie & Rieser, 2022)** The Medical safety dataset is a written English dataset consisting of 2,917 risk-graded medical and non-medical queries (1,417 and 1,500 examples respectively). The dataset was constructed by collecting questions posted on Reddit, such as r/AskDocs. The medical

queries have been labelled by experts and crowd annotators for both relevance and levels of risk (i.e. non-serious, serious to critical) following established World Economic Forum (WEF) risk levels designated for chatbots in healthcare. We merge the medical queries of different risk-levels into one class, given the high scarcity of the latter 2 labels to create an in-domain/out-of-domain classification task for medical queries. Additionally, we consider only the medical queries that were labelled as such by expert medical practitioners. Thus this dataset will facilitate discussion on how to guarantee a system recognises medical queries, to avoid generating medical output.

# G  LiRPA Algorithm Details

Below we briefly present the details of the innerworkings of the auto-LiRPA framework, as outlined in Xu et al. (2020). For more detailed explanations and definitions, please refer to the original paper.

The final goal is to compute provable lower and upper bounds for the value of output node $h_o(\mathbf{X})$, i.e., lower bound $\underline{\mathbf{h}}_o$ and upper bound $\overline{\mathbf{h}}_o$ (element-wise), when $\mathbf{X}$ is perturbed within $\mathbb{S}$: $\underline{\mathbf{h}}_o \leq h_o(\mathbf{X}) \leq \overline{\mathbf{h}}_o$, $\forall \mathbf{X} \in \mathbb{S}$. In LiRPA, we find tight lower and upper bounds by first computing linear bounds w.r.t. $\mathbf{X}$:

$$\underline{\mathbf{W}}_o\mathbf{X} + \underline{\mathbf{b}}_o \leq h_o(\mathbf{X}) \leq \overline{\mathbf{W}}_o\mathbf{X} + \overline{\mathbf{b}}_o \quad \forall \mathbf{X} \in \mathbb{S}, \tag{8}$$

where $h_o(\mathbf{X})$ is bounded by linear functions of $\mathbf{X}$ with parameters $\underline{\mathbf{W}}_o, \underline{\mathbf{b}}_o, \overline{\mathbf{W}}_o, \overline{\mathbf{b}}_o$. We generalize existing LiRPA approaches into two categories: *forward mode* perturbation analysis and *backward mode* perturbation analysis. Both methods aim to obtain bounds equation 8 in different manners:

- **Forward mode**: forward mode LiRPA propagates the linear bounds of each node w.r.t. all the independent nodes, i.e., linear bounds w.r.t. $\mathbf{X}$, to its successor nodes in a forward manner, until reaching the *output node o*.

- **Backward mode**: backward mode LiRPA propagates the linear bounds of *output node o* w.r.t. *dependent nodes* to further predecessor nodes in a backward manner, until reaching all the *independent nodes*.

**Forward Mode LiRPA on General Computation Graphs** For each node $i$ on the graph, we compute the linear bounds of $h_i(\mathbf{X})$ w.r.t. all the independent nodes:

$$\underline{\mathbf{W}}_i\mathbf{X} + \underline{\mathbf{b}}_i \leq h_i(\mathbf{X}) \leq \overline{\mathbf{W}}_i\mathbf{X} + \overline{\mathbf{b}}_i \quad \forall \mathbf{X} \in \mathbb{S}.$$

We start from independent nodes. For an independent node $i$, we have $h_i(\mathbf{X}) = \mathbf{x}_i$ so we trivially have the bounds $\mathbf{I}\mathbf{x}_i \leq h_i(\mathbf{X}) \leq \mathbf{I}\mathbf{x}_i$. For a dependent node $i$, we have a *forward LiRPA oracle function $G_i$* which takes $\underline{\mathbf{W}}_j, \underline{\mathbf{b}}_j, \overline{\mathbf{W}}_j, \overline{\mathbf{b}}_j$ for every $j \in u(i)$ as input and produce new linear bounds for node $i$, assuming all node $j \in u(i)$ have been bounded:

$$(\underline{\mathbf{W}}_i, \underline{\mathbf{b}}_i, \overline{\mathbf{W}}_i, \overline{\mathbf{b}}_i) = G_i(\{B_j | j \in u(i)\}), \text{where } B_j := (\underline{\mathbf{W}}_j, \underline{\mathbf{b}}_j, \overline{\mathbf{W}}_j, \overline{\mathbf{b}}_j). \tag{9}$$

We defer the discussions on oracle function $G_i$ to a Xu et al. (2020). Extending this method to a general graph with known oracle functions, the forward mode perturbation analysis is straightforward to extend to a general computational graph: for each dependent node $i$, we can obtain its bounds by recursively applying equation 9. We check every input node $j$ and compute the bounds of node $j$ if they are unavailable. We then use $G_i$ to obtain the linear bounds of node $i$. The correctness of this procedure is guaranteed by the property of $G_i$: given $B_j$ as inputs, it always produces valid bounds for node $i$.

**Backward Mode LiRPA on General Computation Graphs** For each node $i$, we maintain two attributes: $\underline{\mathbf{A}}_i$ and $\overline{\mathbf{A}}_i$, representing the coefficients in the linear bounds of $h_o(\mathbf{X})$ w.r.t $h_i(\mathbf{X})$:

$$\sum_{i \in \mathbf{V}} \underline{\mathbf{A}}_i h_i(\mathbf{X}) + \underline{\mathbf{d}} \leq h_o(\mathbf{X}) \leq \sum_{i \in \mathbf{V}} \overline{\mathbf{A}}_i h_i(\mathbf{X}) + \overline{\mathbf{d}} \quad \forall \mathbf{X} \in \mathbb{S}, \tag{10}$$

where $\underline{\mathbf{d}}, \overline{\mathbf{d}}$ are bias terms that are maintained in our algorithm. Suppose that the output dimension of node $i$ is $s_i$, then the shape of matrices $\underline{\mathbf{A}}_i$ and $\overline{\mathbf{A}}_i$ is $s_o \times s_i$. Initially, we trivially have

$$\underline{\mathbf{A}}_o = \overline{\mathbf{A}}_o = \mathbf{I}, \quad \underline{\mathbf{A}}_i = \overline{\mathbf{A}}_i = \mathbf{0}(i \neq o), \quad \underline{\mathbf{d}} = \overline{\mathbf{d}} = \mathbf{0}, \tag{11}$$

which makes equation 10 hold true. When node $i$ is a dependent node, we have a *backward LiRPA oracle function* $F_i$ aiming to compute the lower bound of $\underline{\mathbf{A}}_i h_i(\mathbf{X})$ and the upper bound of $\overline{\mathbf{A}}_i h_i(\mathbf{X})$, and represent the bounds with linear functions of its predecessor nodes $u_1(i), u_2(i), \cdots, u_{m(i)}(i)$:

$$(\underline{\boldsymbol{\Lambda}}_{u_1(i)}, \overline{\boldsymbol{\Lambda}}_{u_1(i)}, \underline{\boldsymbol{\Lambda}}_{u_2(i)}, \overline{\boldsymbol{\Lambda}}_{u_2(i)}, \cdots, \underline{\boldsymbol{\Lambda}}_{u_{m(i)}(i)}, \overline{\boldsymbol{\Lambda}}_{u_{m(i)}(i)}, \underline{\boldsymbol{\Delta}}, \overline{\boldsymbol{\Delta}}) = F_i(\underline{\mathbf{A}}_i, \overline{\mathbf{A}}_i),$$

$$\text{s.t.} \quad \sum_{j \in u(i)} \underline{\boldsymbol{\Lambda}}_j h_j(\mathbf{X}) + \underline{\boldsymbol{\Delta}} \le \underline{\mathbf{A}}_i h_i(\mathbf{X}), \;\; \overline{\mathbf{A}}_i h_i(\mathbf{X}) \le \sum_{j \in u(i)} \overline{\boldsymbol{\Lambda}}_j h_j(\mathbf{X}) + \overline{\boldsymbol{\Delta}}. \quad (12)$$

We substitute the $h_i(\mathbf{X})$ terms in equation 10 with the new bounds equation 12, and thereby these terms are backward propagated to the predecessor nodes and replaced by the $h_j(\mathbf{X})(j \in u(i))$ related terms in equation 12. In the end, all such terms are propagated to the independent nodes and $h_o(\mathbf{X})$ will be bounded by linear functions of independent nodes only, where equation 10 becomes equivalent to equation 8.

## LLM USAGE

LLMs were used to aid and polish writing (e.g., grammar and spell checks)

