# OpenReview forum: "CerCE: Towards Certifiable Continual Learning"
_ICLR.cc/2026/Conference — Submitted to ICLR 2026_

### Official Review · Reviewer_8qMj · 2025-10-31

**Soundness:** 3
**Presentation:** 2
**Contribution:** 3
**Rating:** 4
**Confidence:** 3

**Summary:**

This paper proposes CerCE, a continual learning method that provides provable certificates of non-forgetting during training. It treats weight updates as structured perturbations and applies LiRPA to guarantee that past samples remain correctly classified within a small parameter radius. The method is formulated as a constrained optimization problem, solved via gradient projection or a Lagrangian relaxation. Experiments show that the final accuracy is similar to other baselines, while achieving high Average Certification (AC) (a measure of how many stored samples are provably safe from forgetting).

**Strengths:**

1. Introduces a principled framework for provable non-forgetting using LiRPA, which is a novel direction in continual learning.

2. Demonstrates that accuracy can be maintained while adding certifiable safety constraints.

3. Provides empirical validation on both standard and safety-critical datasets such as RUARobot.

**Weaknesses:**

Major concerns:
1. The Average Certification (AC) metric is computed only on the replay buffer used during training (Lines 357-359). This means the certificates guarantee non-forgetting only for those stored samples, not for unseen or full-task data. It also creates a circular evaluation: the model is guaranteed to be "certified" on the very examples it was trained to protect. This limits the interpretability of AC as a genuine measure of generalization or robustness.

2. CerCE achieves final accuracy comparable to rehearsal-based baselines such as ER and DER++ (Table 1). While this parity shows that certification does not harm learning, it also indicates that the method offers no empirical performance gain, with its contribution being primarily theoretical rather than practical.

3. The reported training cost (9 times slowdown stated in Appendix E) may limit scalability to complex architectures and realistic continual learning cases.

**Questions:**

1. Can authors clarify whether the certification guarantees (AC) generalize beyond the replay buffer? In other words, does high AC correlate with actual non-forgetting on unseen or test data?

2. Since CerCE's final accuracy is similar to other baselines, what practical advantages should we expect from adopting it beyond the theoretical certificates?

3. The reported 9 times slowdown is significant. Are there any feasible ways to reduce it for larger models?

---

> ### Author Response · Authors · 2025-11-19
>
> We would like to thank the reviewer for confirming the novelty of our work and the thoroughness of our evaluation.
>
> Regarding the weaknesses and questions posed by the reviewer:
>
> 1. **Generalization Beyond the Buffer:** Firstly, we provide PAC-Bayes generalization bounds (Sec 4.3), bridging the gap between formal guarantees on seen buffer samples and tighter generalization bounds on the test distribution. Second, it is worth noting that the buffer size is a controllable hyperparameter, which can be increased to improve coverage of the test distribution; however, one should note that providing formal guarantees for specific unseen data samples is akin to creating a verifiable “oracle” function, which is extremely difficult, if not impossible, without strong assumptions on the data distribution.
> 2. **Practical Advantages:** It is worth noting that our certificates can be practical in safety-critical applications where measurements beyond accuracy are important to safety. As an example, for applications such as self-driving cars, edge cases such as detecting pedestrians in the rain prove difficult, and certifying the correctness of past samples can be crucial. With that said, as our method is among the first to explore parameter space certificates for continual learning, limitations such as the need for a memory buffer or architectural/computational limitations exist, which can be further improved in future works, resulting in further practical benefits.
> 3. **Computation Cost:** We would like to reiterate that there is a natural cost associated with providing formal certificates. With that said, there are existing methods that improve the speed, precision, and large-scale compatibility of calculating LiRPA bounds \[1\]. Because the general principles remain the same as in the seminal works cited in our paper, we do expect that advances in the certification literature could be applied to our framework. There would be some engineering effort required, as these advances are usually focused on problems where uncertainty is at the network input, as opposed to uncertain parameters.
>
> \[1\] Shi, Zhouxing, Qirui Jin, Zico Kolter, Suman Jana, Cho-Jui Hsieh, and Huan Zhang. "Neural network verification with branch-and-bound for general nonlinearities." In International Conference on Tools and Algorithms for the Construction and Analysis of Systems, pp. 315-335. Cham: Springer Nature Switzerland, 2025\.

---

> ### Comment · Reviewer_8qMj · 2025-11-26
>
> Thank the authors for the clarifications. I think the answers did not directly address my concerns. I will keep the rating at this moment, and appreciate if the authors can clarify and answer the previous concerns directly
>
> Specifically, the first concern is whether the AC is evaluated *only** on the *training* buffer, not the test dataset. This relates to whether the AC can be generalized beyond the training data buffer to test data within the same task.
>
> I appreciate that authors discuss the practical advantages in 'real application', but I hope to clarify that the second concern was why the final accuracy is comparable to the rehearsal-based baseline. If the proposed method is effective, the final average accuracy of both *previous tasks* and the current task is expected to be improved.
>
> For the third part, I agree that the increased cost relates to the literature of formal certificates. However, this seems to exactly prove the weakness that including certification literature brings only theoretical benefits, but induces high practical costs (9 times slowdown stated in Appendix E). One direct evidence would be better performance under the *same* computational cost.

---

> > ### Author Response · Authors · 2025-11-26
> >
> > We appreciate the reviewer clarifying their questions. We would like to respond further below:
> >
> > >Specifically, the first concern is whether the AC is evaluated only* on the training buffer, not the test dataset. This relates to whether the AC can be generalized beyond the training data buffer to test data within the same task.
> >
> > - Firstly, we believe our initial answer does directly address generalization to test data, as that is the purpose of a generalization bound. Furthermore, we conducted additional experiments regarding the empirical correlation between buffer certification and the test set accuracy of each class, after each task. The results are presented in **Fig. 3** of the revised manuscript and indicate a positive correlation between per-class **buffer certification ratio** and **test-set accuracy**.
> >
> > >I appreciate that authors discuss the practical advantages in 'real application', but I hope to clarify that the second concern was why the final accuracy is comparable to the rehearsal-based baseline. If the proposed method is effective, the final average accuracy of both previous tasks and the current task is expected to be improved.
> >
> > - As the reviewer mentions, it is true that certification doesn't hurt performance. But note that the main goal of certificates is **not** empirical performance gain. As we mentioned in our previous response, there are cases in which "measurements beyond accuracy are important to safety". While we do not necessarily improve task accuracy, we can provide guarantees about specific scenarios important to the application (see initial response for examples).
> >
> > - Additionally, one should note that certified training methods often result in severe degradations of performance, as previously observed in the adversarial robustness literature [1]. We believe that beyond the practical scenarios mentioned in our initial response, the fact that CerCE attains competitive accuracy with rehearsal-based methods *while providing certificates* is a practical advantage.
> >
> >
> >
> > > For the third part, I agree that the increased cost relates to the literature of formal certificates. However, this seems to exactly prove the weakness that including certification literature brings only theoretical benefits, but induces high practical costs (9 times slowdown stated in Appendix E). One direct evidence would be better performance under the same computational cost.
> >
> > - In our initial response, we referred to existing works that can be utilized to reduce the computational costs of formal certificates, and we have acknowledged the limitations of our method in the conclusion section of the revised manuscript.
> >
> > - Furthermore, the reviewers comment that formal certification *"brings only theoretical benefits"* follows an implicit assumption that said theoretical benefits are not *practical*. We would like to emphasize once again that formal guarantees are invaluable in heavily regulated and safety-critical industries such as banking, autonomous driving, and healthcare. Certificates, on their own, are helpful. Not just as a tool to improve accuracy.
> >
> >
> > [1] Mao, Yuhao, et al. "Connecting certified and adversarial training." Advances in Neural Information Processing Systems 36 (2023): 73422-73440.

---

### Official Review · Reviewer_sy7A · 2025-10-31

**Soundness:** 3
**Presentation:** 2
**Contribution:** 3
**Rating:** 6
**Confidence:** 5

**Summary:**

This paper introduces Certifiable Continual Learning (CerCE), a novel framework that provides formal guarantees against catastrophic forgetting in continual learning by leveraging Linear Relaxation Perturbation Analysis (LiRPA). CerCE reinterprets weight updates as structured perturbations, deriving constraints to ensure the preservation of past knowledge and formulating continual learning as a constrained optimization problem. The framework employs practical strategies like gradient projection and Lagrangian relaxation and connects to PAC-Bayesian generalization theory, leading to tighter generalization bounds and reduced memory overfitting. Experimental results on various datasets showcase CerCE's strong empirical performance and its unique ability to offer theoretical guarantees for knowledge retention, paving the way for verifiable continual learning in safety-critical applications.

**Strengths:**

- A novel approach that attempts to establish "guardrails" for continual learning (CL), with potentially great impact in safety-critical applications.
- The continual learning field often lacks standardization and formal approaches, relying heavily on heuristics. The framework introduced by the authors is refreshing and has the potential to inspire new avenues for formal methods in CL.
- I am not an expert in PAC-Bayes Bounds, but the authors are able to tighten a classic bound by theoretically guaranteeing performance on a buffer memory set.

**Weaknesses:**

- I am not entirely comfortable with claims of "theoretical guarantees" paired with "linear relaxation". I would rather describe these as theoretically grounded certificates. The tighter PAC-Bayes bound also relies on the assumption that this relaxation formally guarantees performance on the buffer set (or buffer minibatch), but whether this holds true in general is not clear to me.
- I believe more could be done to connect certificate satisfaction and backward generalization in order to validate the approach.
- Applicability is tied to architectures where LiRPA is computable. Not being familiar with LiRPA, I question the scalability of such an approach with respect to architectural complexity.

**Questions:**

1. Exposition could be improved:
    - The notation in Theorem 1 and Corollary 1.1 could be clearer. I had to consult the appendix and reference works to fully understand it. Even simply introducing the dimensionality of the bound parameters could aid readability.
    - Figure 2: The y-axis of the second plot is missing.
    - Ablation 5.2: "Ratio of Certified Samples in the Buffer" does not seem like an ablation study; I suggest moving it to the experimental section or appendix.


2. Questions:
    - Table 3: If the "bound" case includes all samples that satisfy the constraint, wouldn't that imply an Average Certification (AC) of 100%? Are you computing AC over the entire buffer instead of just the buffer sample?
    - Following the corollary on the PAC-Bayes bound, you state that it leads to less overfitting on the memory buffer. While an argument for better "backward generalization capabilities" can be made, the implication regarding buffer memory overfitting is less clear to me (especially since in Corollary 2.1 you assume that `\max_{\|\Delta \theta\|_2}\hat{l}_\mathcal{M}(\theta)=0`).
    - Could this method actually be used to train with guarantees of non-forgetting in practical scenarios, or is it currently more of a theoretical possibility? What are the limitations in this regard?
    - It would significantly increase the soundness of the paper if you empirically demonstrate the connection between certificate satisfaction and per-task backward generalization (BG) as new tasks are learned. Does task performance retention scale with the satisfied certificates per class?
    - It would be reassuring to mention the limits of this methodology (if any), such as its scaling with respect to network depth or general issues with the approximation in the weights space.


Satisfactory answers to these points could lead me to re-evaluate my grade, and I believe it would significantly strengthen the work. I'm looking forward to the discussion, and I encourage the authors to continue this interesting line of research.

---

> ### Author Response · Authors · 2025-11-19
> **Response No. 1**
>
> We would first like to thank the reviewer for acknowledging the novelty of our work and recognizing our effort to inspire formal approaches to CL, which is lacking in the current literature.
> Regarding the weaknesses mentioned by the reviewer:
>
> 1. **Linear Bounds:** It is important to note that we are not doing any *linearization*; we are doing a *linear relaxation*, which is a convex relaxation, which considers every set of model parameters within a domain, thus our theoretical guarantees are sound. That is, the lower and upper linear bounds are abstractions rather than approximations, with proofs and details mentioned in our paper as well as cited seminal works.
> 2. **Generalization Experiments:** We have conducted additional experiments in this regard, as discussed below in response to question 4\.
> 3. **Architectural Compatibility:** We would first like to highlight that our theory and methodology are not limited by architectures. Additionally, the current limitations regarding complex architecture are due to the fact that we are among the first and few works that utilize the parameter-space verification aspect of LiRPA, which is currently underdeveloped. However, even with the current limitations, we expect our method could be extended to more complex architectures through parameter-efficient fine-tuning methods such as Low Rank Adaptation (LoRA), as a LoRA adapter is a linear function and would be supported by the weight perturbation analysis of LiRPA. Additionally, because the general principles remain the same as in the seminal works cited in our paper, we do expect that advances in the certification literature could be applied to our framework.
>
> Regarding the exposition:
>
> 1. **LiRPA coefficient dimensions:** We have added the dimensionalities of the coefficients in the line following Theorem 1 to improve clarity. We have also corrected some slight notational issues with the Lagrangian formulation as a result.
> 2. **Axis numbers:** The numbers on the axes were omitted due to the two plots sharing the y-axis. We have added the numbers back to the right plot to improve readability.
> 3. **Section Title:** We have changed the name of the section to “Ablation Studies and Exploratory Experiments” to better reflect the nature of the content.
>
> In order to conform to the OpenReview character limit for each comment, we will respond to the reviewer's questions in the comment below.

---

> > ### Author Response · Authors · 2025-11-19
> > **Response No. 2**
> >
> > (This is a continuation of the previous comment)
> >
> > Regarding the reviewer’s questions:
> >
> > 1. **Table 3 Bound Scenario:** As mentioned in Section 5, AC is indeed defined and measured over the entire buffer. It is also noteworthy that in the “bound” scenario, while a sample is certified when it is sampled into the buffer, this certification can be violated during later training steps, and the purpose of the objective function is to reduce this occurrence as much as possible.
> > 2. **Corollary 2.1 and Backward Generalization:** We would like to note that less “memory overfitting” is synonymous with more “backward generalization,” as the memory buffer is representative of previous task data, and the concepts of “overfitting” and “generalization” are opposite each other. Regarding  Corollary 2.1 and the generalization bounds, we would like to reiterate that our linear bounds are not approximations, and if the constraints provided by LiRPA are satisfied for the considered samples, the assumption in Corollary 2.1 holds, resulting in a tighter upper bound on the error on the test distribution, meaning better “backward generalization.”
> > 3. **Practicality of CerCE:** It is worth noting that our certificates can be practical in safety-critical applications where measurements beyond accuracy are important to safety. As an example, for applications such as self-driving cars, edge cases such as detecting pedestrians in the rain prove difficult, and certifying the correctness of past samples can be crucial. With that said, as our method is among the first to explore parameter space certificates for continual learning, limitations such as the need for a memory buffer or architectural/computational limitations exist, which can be further improved in future works.
> > 4. **Per-class Certification and Past Task Test Accuracy:** We have conducted an experiment as suggested, investigating the relationship of the per-class certification ratio with test set accuracy. We present the results in the newly added Fig. 3 of the revised manuscript, and we observed that there is indeed a strong correlation between the two aforementioned metrics. Additionally, we conduct a similar experiment and discussion while using ER as the training method in Fig. 6 of Appendix D, and to summarize, the results are consistent with our observations in Fig. 3, even though the certification rates are low for ER, there is still a positive correlation with accuracy. We believe this evidence demonstrates the positive impact of certificates on backward generalization as suggested by the PAC-Bayes bounds.
> > 5. **Limitations:** As CerCE is the first to pursue certifiable continual learning through linear relaxations, there are limitations such as architectural compatibility, additional computational overhead, and reliance on a memory buffer. However, all of the above-mentioned limitations are not inherent to CerCE and can be improved with further development of faster and more compatible LiRPA algorithms that can calculate tighter upper and lower bounds, which are valid over larger radii in the parameter space. CerCE provides a necessary first step for future work to extend and improve certification in continual learning, which is necessary for deployment in real-world safety-critical systems. We have added this discussion of limitations to the conclusion of the paper.

---

### Official Review · Reviewer_ooBX · 2025-11-03

**Soundness:** 3
**Presentation:** 2
**Contribution:** 3
**Rating:** 4
**Confidence:** 4

**Summary:**

This paper introduces Certifiable Continual LEarning (CerCE), a framework designed to address catastrophic forgetting in continual learning (CL) by providing formal guarantees of non-forgetting. The authors utilize Linear Relaxation Perturbation Analysis (LiRPA) to interpret weight updates as structured perturbations, thereby deriving constraints that aim to preserve previously acquired knowledge. The method is formulated as a constrained optimization problem and evaluated on standard CL benchmarks as well as safety-critical datasets.

**Strengths:**

Adding formal guarantees to continual learning is particularly relevant for the deployment of CL systems in safety-critical applications (a domain appropriately explored in the experiments). The core idea of leveraging neural network certification techniques (specifically LiRPA) to define constraints on parameter updates is innovative and presents a novel angle in the field, as far as I know. The concept of analyzing parameter permutations that lead to changes in accuracy (the "forgetting-set") is powerful and opens up a potentially rich area for future research, bridging the gap between the certification and CL literature.

**Weaknesses:**

While the core idea is promising, the paper in its current form feels underdeveloped and would benefit significantly from a deeper analysis of the proposed mechanisms. The authors introduce several interesting concepts (e.g., LiRPA coefficients, certification constraints) but stop short of exploring their full implications.

The main weaknesses are:

* **Insufficient Analysis:** The paper would be much stronger if it focused more on the ablation and analysis of the approach. For example, a deeper discussion of what the LiRPA coefficients specifically capture about the data or the parameter space would be highly valuable. It is also unclear how these certification-based constraints relate to existing approaches that impose constraints on parameter updates in CL (e.g., projection-based methods).
* **Unfair Complexity Comparison:** The experimental comparison to baselines (e.g., ER) appears unfair, as the proposed method introduces significant computational overhead. The paper requires a principled discussion of the trade-off between this extra complexity and performance. An analysis of the computational complexity *per buffer sample* for CerCE versus baselines, followed by a comparison of performance at equivalent complexity budgets, would be necessary to fairly situate the method's contributions.
* **Misplaced Theoretical Focus:** The discussion on PAC-Bayesian bounds seems peripheral to the paper's central claim of certification. This section does not add significant value and feels disconnected from the primary certification objective. This space might be better utilized for the deeper analysis mentioned above, such as exploring the relationship between the constraints and the properties of the loss landscape (e.g., flatness).
* **Lack of Experimental Clarity:** The experimental setup description is vague. The authors state they use MLPs on top of pre-trained architectures, but the exact details of these architectures and the complexity of the trained MLPs are missing, making replication difficult.
* **Formatting and Presentation:** The paper suffers from numerous formatting and stylistic issues, which detract from the main message. Figure 2 is of poor quality, table formatting is inconsistent (e.g., Table 1), and the appendix is minimal and poorly structured. For example, Appendix F separates standard deviations into their own table and confusingly labels them "error bars," which is highly unconventional. Phrasing such as "What goes on in the loss" is not appropriate for a formal publication.

**Questions:**

1.  Could the authors clarify how tight the certification guarantees are in practice? For instance, if a sample breaks the certification guarantee during an update, how often does this correlate with the sample actually being misclassified (i.e., forgotten)?
2.  Many results (e.g., average certification) are reported on the replay buffer. Given that the buffer is only a small subset of past data, could the authors comment on why certification on the buffer is the primary metric, rather than guarantees that might apply to the true (unseen) data distribution of past tasks?
3.  Is there a direct, empirically verifiable connection showing that the specific samples *most likely* to be forgotten (e.g., those near a decision boundary) are the ones primarily protected by the certification constraints?
4.  To better contextualize the trade-off between standard accuracy and certification, it would be beneficial to see a visualization of the accuracy drop on *previous tasks' test sets* (not the buffer) after each new task is learned, for all compared methods.
5.  The LiRPA method is central to the paper but is not introduced with sufficient intuition in the main text. Could the authors provide a more self-contained, high-level overview of what these coefficients represent in the main paper?
6.  The cited literature on NN certification appears to be from 2021 or older. Could the authors comment on whether more recent developments in LiRPA or other certification methods might be applicable or simplify their framework?

---

> ### Author Response · Authors · 2025-11-19
> **Response No. 1**
>
> We would like to thank the reviewer for acknowledging the novelty of our work, as well as the importance of formal guarantees in continual learning (CL) for safety-critical systems, which is an underexplored area of research.
> Below, we would like to address the mentioned weaknesses:
>
> 1. **Experimental Thoroughness:** We would like to mention that we’ve performed extensive ablation of the components of our method, with different values of $\\gamma,\\lambda$, buffer sizes, and buffer selection strategies. With regard to further exploration of the coefficients, we have added additional experiments in Section 5 and Appendix D of the revised manuscript, in response to the reviewers’ comments, which we discuss individually in the questions section below.
> 2. **Computation Cost:** We would like to emphasize that CerCE is among the first to offer formal guarantees to CL, which naturally incurs some computational costs. Most competing methods do not provide certificates, and the only existing work that attempts to provide guarantees, InteContiNet, incurs similar slowdowns to CerCE despite worse performance overall. (See Table 8.)
> 3. **Theoretical Generalization Bounds:** We believe the connection between the provided PAC-Bayes bound and the main method is well justified; the PAC-Bayes bound connects the certificates provided on seen buffer data samples to generalization bounds on future unseen data. In fact, we believe the PAC-Bayes bound is precisely the answer to the reviewer’s second question regarding generalization to unseen data, which we discuss in the response to question 2 below.
> 4. **Architecture Details:** In Section 5, we have highlighted the pre-trained architectures used with proper citation to the original works. We have now included the missing details in the revised manuscript. Regarding the ViT size, we use ViT-B, and the MLP has 3 layers, 400 hidden dimensions, and ReLU activation across all experiments. Our original submission included the source code as supplementary material to ensure full experimental transparency and reproducibility.
> 5. **Formatting and Presentation:** We thank the reviewer for pointing out the issue with the phrasing in “What goes in the loss” and have corrected it to more formal language to “*Buffer and Loss Function Sample Selection”*. Tables 1 & 2 follow the same formatting. Figure 2 is embedded as a PDF using vector graphics, meaning there are no resolution issues. Perhaps the left subfigure is confusing to the reviewer due to the overlap of the two curves. We believe the length of the appendix (6 pages, in our case) has no bearing on the quality of our work. Representing “error bars” as one standard deviation is a common practice \[1\]. We have now moved the error bars next to the main values in tables 1 & 2 in the revised manuscript.
>
> In order to conform to the OpenReview character limit for each comment, we will respond to the reviewer's questions in the comment below.

---

> > ### Author Response · Authors · 2025-11-19
> > **Response No. 2**
> >
> > (This is a continuation of the previous comment)
> >
> > Regarding the reviewer’s questions:
> >
> > 1. **Certification and Accuracy Correlation:** When a sample in the buffer violates the constraints, we do not observe a direct correlation between *that specific sample* being misclassified, as most of the buffer samples are classified correctly throughout training, as seen in Fig. 2\. However, in the newly added Fig. 3 of the revised manuscript, we observe that there is a positive correlation between per-class certification ratio *in the buffer* and prediction accuracy *in the test set.*
> > 2. **PAC-Bayes bound importance:** We would like to reiterate our response from the weaknesses section. The provided PAC-Bayes bound bridges the gap between providing certificates on seen buffer data and generalization guarantees to unseen data. Note that providing formal guarantees for specific unseen data samples is akin to creating a verifiable “oracle” function, which is extremely difficult, if not impossible, without strong assumptions on the data distribution. Furthermore, CerCE does not significantly sacrifice other metrics, such as accuracy, for the sake of providing certificates (as seen in Tables 1 & 2).
> > 3. **Saved Samples and Decision Boundary:** As the decision boundary changes incrementally during gradient descent, we would expect the samples protected by CerCE to be generally closer to the decision boundary. However, while this is difficult to verify for high-dimensional data, as the decision boundary is both difficult to compute as well as difficult to plot, we investigated this interesting question from the reviewer by conducting an experiment for our 2D toy example in Fig. 5 of the revised manuscript (Appendix A). In this new experiment, we highlight the samples that would be forgotten if a naive SGD approach were used instead of CerCE for each epoch, and observe that they are indeed closer to the decision boundary.
> > 4. **Past Task Accuracy:** While one of our primary metrics, Final Accuracy, incorporates past task test accuracy, we have provided additional task-specific test accuracies over time at the request of the reviewer in the appendix section D of the revised manuscript (Figure 7 & Figure 8). We observe that, consistent with Table 1, CerCE is able to maintain competitive previous task accuracy to other buffer-based methods, while providing certificates. Additionally, CerCE generally performs better on retaining past-task performance rather than improving newer task accuracies, which is in line with our intuition since the focus of CerCE is on providing formal certificates for past data samples.
> > 5. **LiRPA explanation:** We would like to note that the paper discusses LiRPA methods in detail in Section 2, Section 4, and with further detail in Section B of the appendix. However, for improved clarity, we have added additional high level explanations to Section 2 of the revised manuscript: “At a high level, given a set of initial permissible perturbations to a neural network input or parameters, LiRPA offers formal upper and lower bounds to the network output in the form of *linear functions of the input or parameters.* It is worth noting that these bounds are not approximations, and are rather abstractions, and do hold in practice.” Additionally, we have added further details to Theorem 1 to include the dimensionality of the LiRPA coefficients for better clarity.
> > 6. **LiRPA Developments:** More recent developments in LiRPA, such as leveraging branch-and-bound \[2\], can provide tighter bounds, scale to larger networks, and offer major computational speedups. Because the general principles remain the same as in the seminal works cited in our paper, we do expect that advances in the certification literature could be applied to our framework. There would be some engineering effort required, as these advances are usually focused on problems where uncertainty is at the network input, as opposed to uncertain parameters.
> >
> >
> > \[1\] [https://en.wikipedia.org/wiki/Error\_bar](https://en.wikipedia.org/wiki/Error_bar)
> >
> > \[2\] Shi, Zhouxing, Qirui Jin, Zico Kolter, Suman Jana, Cho-Jui Hsieh, and Huan Zhang. "Neural network verification with branch-and-bound for general nonlinearities." In International Conference on Tools and Algorithms for the Construction and Analysis of Systems, pp. 315-335. Cham: Springer Nature Switzerland, 2025\.

---

### Author Response · Authors · 2025-12-02
**Final Comment by Authors**

We would like to extend our gratitude to the ICLR Reviewers, ACs, and PCs for their efforts. We especially appreciate the AC’s commitment, given the circumstances. Below, we would like to provide a final summary of our contributions, the reviews, our responses, and the brief discussion period.

**Our contributions:**
In this work, we introduced Certifiable Continual Learning (CerCE), a novel framework that brings formal guarantees to the problem of continual learning (CL), an area that is severely unexplored in the CL literature. Our method not only addresses certifiable continual learning but also improves generalization by tightening PAC-Bayes bounds, helping mitigate memory overfitting, a persistent challenge in rehearsal-based methods.

Below is the summary regarding reviewers’ scores:

- **Reviewer sy7A:** 6 $\\rightarrow$ 8(?) (Indicated *“Satisfactory answers to these points could lead me to re-evaluate my grade”* and *“I encourage the authors to continue this interesting line of research”*, could not respond before the end of the discussion)
- **Reviewer ooBX:** 4 (No response before the end of the discussion)
- **Reviewer 8qMj:** 4 (Asked for further clarification, could not respond further before the end of the discussion)


Main changes to the manuscript and response to the main concerns:

- **Buffer Certification and Generalization:** Beyond our proven tighter generalization bound, we conducted additional experiments demonstrating the positive empirical connection between buffer certification and backward generalization, answering the reviewers' concerns regarding certificates belonging to the buffer.
- **Practicality of Certificates:** We emphasized that formal certificates on their own are integral to applications that are concerned with safety and transparency, and are not just a means to increase accuracy.
- **Computational Complexity and Scaling Issues:** We emphasize that while our method comes with additional computational costs, it is able to maintain competitive empirical performance with state-of-the-art, while offering formal certificates, a significant advantage for any certification method. Furthermore, we pointed to existing works that can be incorporated to improve the efficiency and compatibility of our method.
- **Boundary Sample Analysis:** We conducted an experiment on a toy example, suggesting that CerCE certifications contribute to protecting the samples closest to the decision boundary.
- **Past Task Accuracy:** We included plots indicating the per-task accuracy during training for additional experimental transparency.
- **Limitations Discussion:** We included the discussion of limitations in computational complexity and architecture compatibility brought by the reviewers in the main text. Additionally, we emphasize that the limitations are due to the fact that we are among the first and few works to utilize weight-perturbation-based certification, an area that is currently underdeveloped, and we hope that our contribution can ignite future improvements.
- **Additional Exposition and Notational Issues:** We added additional explanations regarding the details of LiRPA and our method, improved table and error bar formatting, and fixed several notational issues. Furthermore, we emphasized that our certifications are not approximations and are formally sound.

**Summary:** Theoretically verifiable guarantees, i.e., certificates, are an invaluable asset in safety-critical and heavily regulated applications of machine learning. The existing continual learning literature has seldom explored this area; To the best of our knowledge, only one existing work touches on providing guarantees, leading to suboptimal performance as discussed in our paper. In contrast, CerCE provides certificates while maintaining similar performance to existing methods. Being one of the first of its kind, our method comes with additional computational costs and scaling limitations; however, said limitations are not inherent to our methodology and rather a symptom of the underdevelopment of certification tools and frameworks. We believe that our contribution is a significant step towards safer and more transparent machine learning hope that this paper inspires future works to advance the endeavor.

---

### Meta-Review · Area_Chair_GZ1N · 2026-01-10

**Summary:**

This paper introduces Certifiable Continual LEarning (CerCE), which leverages Linear Relaxation Perturbation Analysis to interpret weight updates as structured perturbations. The method is formulated as a constrained optimization problem, solved via gradient projection or a Lagrangian relaxation.

One reviewer finds the work technically sound and conceptually novel. In contrast, one reviewer raises substantial concerns regarding the empirical evaluation, particularly the reliance on buffer-level certification and the lack of direct evidence of generalization to unseen data, while another reviewer strongly questions the practical value of the proposed approach given its significant computational overhead. The rebuttal is detailed and technically thorough, addressing many concrete theoretical and experimental concerns. However, there are still several concerns not been addressed.

**Reviewer Concerns:**

A central unresolved weakness across multiple reviews is that the Average Certification metric is evaluated exclusively on the replay buffer. While the authors argue that PAC-Bayesian bounds and empirical correlations provide indirect evidence of generalization, no direct certification is provided on held-out test data.

In addition, while the authors emphasize that certification is not intended to improve accuracy, several reviewers found it difficult to justify the substantial computational overhead in the absence of either superior empirical performance, or stronger empirical evidence that the certificates deliver practical benefits beyond theoretical guarantees.

After reading the paper and rebuttal, I think these two major weaknesses are still not addressed by the rebuttal. I note the performance in Table 1 is better than the baseline, but the current submission does not yet provide sufficiently strong empirical evidence or evaluation to support the balance between the certification, scalability and computational cost.

**Reviewer Scores:**

Reviewer ooBX: Rated the paper slightly below the acceptance threshold. However, the cost-performance trade-off and practical utility of the method remain unclear.

Reviewer sy7A: Rated the paper slightly above the acceptance threshold. The rebuttal adequately addressed the raised concerns.

Reviewer 8qMj:  Rated the paper slightly below the acceptance threshold and retained the original score, citing the lack of empirical performance gains over rehearsal-based baselines and the significant computational overhead weakens the practical motivation. Based on comments, the score is unlikely to change.

---

### Decision · Program_Chairs · 2026-01-26

Reject